# Not too little, not too much: a theoretical analysis of graph (over)smoothing

**Nicolas Keriven**
CNRS, GIPSA-lab, Grenoble, France
`nicolas.keriven@cnrs.fr`

## Abstract

We analyze graph smoothing with *mean aggregation*, where each node successively receives the average of the features of its neighbors. Indeed, it has quickly been observed that Graph Neural Networks (GNNs), which generally follow some variant of Message-Passing (MP) with repeated aggregation, may be subject to the *oversmoothing* phenomenon: by performing too many rounds of MP, the node features tend to converge to a non-informative limit. In the case of mean aggregation, for connected graphs, the node features become constant across the whole graph. At the other end of the spectrum, it is intuitively obvious that *some* MP rounds are necessary, but existing analyses do not exhibit both phenomena at once: beneficial "finite" smoothing and oversmoothing in the limit. In this paper, we consider simplified linear GNNs, and rigorously analyze two examples for which a finite number of mean aggregation steps provably improves the learning performance, before oversmoothing kicks in. We consider a latent space random graph model, where node features are partial observations of the latent variables and the graph contains pairwise relationships between them. We show that graph smoothing restores some of the lost information, up to a certain point, by two phenomena: graph smoothing shrinks non-principal directions in the data faster than principal ones, which is useful for regression, and shrinks nodes within communities faster than they collapse together, which improves classification.

## 1 Introduction

In recent years, deep architectures such as Graph Neural Networks (GNNs), along with the availability of large sets of graph data, have significantly broadened the field of machine learning on graphs and structured data, with a myriad of applications ranging from community detection [11] to molecule classification [20], drug discovery [19], quantum chemistry [15], recommender systems [44], semi-supervised learning, and so on. See [7, 16, 6, 46] for reviews. Most GNNs rely on the **Message-Passing** (MP) framework [15, 23], with a plethora of variants. At each layer $k$, for each node $i$, a representation $z_i^{(k)}$ is computed using the representations of the *neighbors* $\mathcal{N}_i$ of $i$ in the graph at the previous layer:

$$z_i^{(k)} = \text{AGG}\left(\{z_j^{(k-1)}\}_{j \in \mathcal{N}_i}\right) \tag{1}$$

where AGG is an **aggregation function** that, crucially, treats $\{z_j^{(k-1)}\}_{j \in \mathcal{N}_i}$ as an *unordered set*, to respect the absence of node ordering in the graph. There are many variants of aggregation functions, based on sum, mean, max, min, degree-normalized [23], attention-based [39], and so on. In this work, we consider one of the most classical, *mean aggregation*:

$$z_i^{(k)} = \frac{1}{\sum_j a_{ij}} \sum_j a_{ij} \Psi\left(z_j^{(k-1)}\right) \tag{2}$$

36th Conference on Neural Information Processing Systems (NeurIPS 2022).

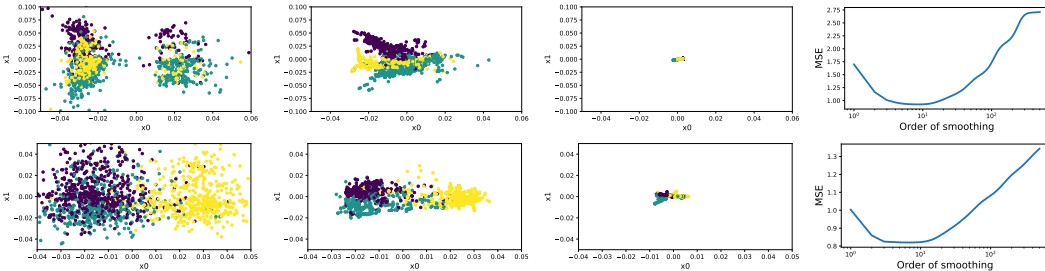

Figure 1: Illustration of both beneficial smoothing and oversmoothing on Cora [32] (top) and Citeseer [14] (bottom). **From left to right:** node features after performing respectively $k = 0, 10$, and $500$ steps of mean aggregation, along the first two principal-components (of the original unsmoothed features), for three classes of nodes for better visibility. **Figure on the right:** Mean Square Error of Linear Ridge Regression (LRR) on the smoothed features with respect to the order of smoothing $k$. We observe that smoothing first gather same-labels nodes and improves learning, before they eventually collapses to a single point (note that here we show LRR for consistency with the analysis presented in this paper, even though these are node classification tasks).

where the $a_{ij} \in \mathbb{R}_+$ are the entries of the adjacency matrix of the graph: either positive edge weights or $0, 1$ for unweighted edges, and $\Psi$ is some function (usually a Multi-Layer Perceptron). In other words, the aggregation process is a weighted average over the neighbors. As we will see, it corresponds to a multiplication by (identity minus) the *random walk Laplacian* of the graph.

While MP is a natural and rather general framework, its limitations were quickly observed by researchers and practitioners. Foremost among them is the so-called *oversmoothing* phenomenon [27]: as the GNN gets deeper and many rounds of MP are performed, the node features $z_i^{(k)}$ tend to become too similar across the graph, especially for small-world graphs with small diameter. Oversmoothing prevents GNN from being too deep unless one is particularly careful. A non-negligible part of the literature is dedicated to fighting oversmoothing with various strategies (see below).

On the theoretical side, oversmoothing has mostly been analyzed in the infinite-layer limit $k \to \infty$. In this case, classical spectral analysis of graph operators such as the Laplacian can be leveraged to indeed show that node features will always converge to some limit that carries a limited amount of information [34]. This is particularly true for mean aggregation (2), with *a constant limit across all nodes* for a connected graph, see Sec. 3. Unlike some other graph operators such as the symmetric normalized Laplacian, where the limit still carries a small amount of information such as the degrees, with the random walk Laplacian *all information* is lost in the limit (beyond a single constant).

However, there has been little research at the other end of the spectrum, showing that *some smoothing is useful for learning*, despite this fact being intuitively and empirically obvious. Generally, researchers show the power of GNNs for a *sufficient* (unbounded) number of layers, such as the now-famous ability to distinguish graph isomorphism as well as the Weisfeiler-Lehman test and all its variants [47, 30], the ability to compute some graph functions [28], and so on. Since these results are valid for an unbounded number of layers, the settings adopted in these works are, by definition, incompatible with non-informative oversmoothing. To our knowledge, there is no work that formally **models both phenomena at once**: *some* smoothing is provably useful for learning, while *too much* smoothing inevitably leads to oversmoothing.

This work aims to fill this gap. We showcase two representative exemples, of regression and classification, on which *linear* GNNs (aka, here, simply Linear Ridge Regression (LRR) on smoothed features) are subject to this double phenomenon. Note that restricting ourselves to *mean aggregation* makes this claim quite non-trivial: in the absence of any "informative" node features, no information can be recovered by mean aggregation alone. For instance, it leaves constant node features unchanged, and the limit $k \to \infty$ is always a constant. So the challenge is the following: node features must carry *some* information, such that a finite number of steps of mean aggregation *provably increases the amount of useful information*, before it loses it in the limit. See Fig. 1 for an illustration.

To show this we adopt on a model of latent space random graphs, with node features. The latter contain partial information about the unobserved latent variables on which both the labels and the graph structure depend. On our examples, we prove that with high probability, graph smoothing improves performance before oversmoothing occurs. We identify two key phenomena for this:

smoothing shrinks non-principal directions in the data faster than principal ones (Sec. 4), and shrinks communities faster than they collapse together (Sec. 5). Although our theoretical settings are obviously simplified, we believe it is a step towards a better comprehension of graph aggregation and of the relationship between node features and graph structure, at the heart of many phenomena in graph machine learning.

**Related Work** Oversmoothing [27] is a very active area of research in geometric deep learning, and an exhaustive list of works would be out of scope here. The research has been mainly focused on novel architectures to relieve it, such as residual mechanisms [26, 10], randomly dropping connections [18], introducing local jumps [48], clever normalizations [50, 17, 5, 37] or regularizations [9], among others. Some works have acknowledged the important role of the aggregation function, and proposed new exotic diffusion strategies [5] or to optimize it [24]. On the theoretical side, it has been mainly shown that repeatedly applying graph smoothing operators indeed induces convergence of the node features [34]. In this work, we analyze a model that present both the benefits of finite smoothing despite oversmoothing in the limit.

Our theoretical framework is based on simplified *linear* GNNs and random graphs that explicitly model the dependence between labels, node features, and graph structure. Despite their simplicity, linear GNNs, sometimes called Simplified Graph Convolutional networks (SGC), have been observed to exhibits relatively good performance [45, 33] and are routinely used in theoretical analyses [51]. Random graphs have been used extensively to analyze graph machine learning algorithms [43, 35] and the theoretical properties of GNNs such as stability [21, 36], transferability [25] or universality [22]. Our model crucially includes observed node features, an essential part in analyzing the smoothing process. They have been shown to be correlated to sought-for labels in real graphs [13], and that this fact is key in the success of GNNs. Our proof is in fact more akin to analyzing a graph diffusion process [31]: given appropriate initial conditions (observed node features), at initial time the diffusion produces a better signal for learning, before it eventually collapses to a single point. To the best of our knowledge, this is the first proof of this kind in a machine learning context.

**Outline** We describe our framework in Sec. 2. In Sec. 3, we briefly prove the oversmoothing phenomenon when $k \to \infty$, which is just the Markov chains ergodic theorem in our settings. In Sec. 4, we study a regression problem. We derive an expression that predicts with good accuracy the optimal smoothing order $k^\star$ in some cases. In Sec. 5, we study a classification problem between two Gaussians. Although we formally prove the *existence* of $k^\star > 0$, deriving an explicit expression for the risk is still open in this case. Code to reproduce the figures is available at `https://github.com/nkeriven/graphsmoothing`.

## 2 Preliminaries

**Notations.** The norm $\|\cdot\|$ is the Euclidean norm for vectors and spectral norm for (rectangular) matrices. For a psd matrix $\Sigma$, the Mahalanobis norm is $\|x\|_\Sigma^2 \stackrel{\text{def.}}{=} x^\top \Sigma x$. The determinant of a matrix is $|S|$, and its smallest eigenvalue is $\lambda_{\min}(S)$. The multivariate Gaussian distribution with mean $\mu$ and covariance $\Sigma$ is denoted by $\mathcal{N}_{\mu,\Sigma}(x) = \det(2\pi\Sigma)^{-\frac{1}{2}} e^{-\frac{1}{2}\|x-\mu\|_{\Sigma^{-1}}^2}$. We will use the shortened notations $\mathcal{N}_\mu = \mathcal{N}_{\mu,\mathrm{Id}}$ and $\mathcal{N} = \mathcal{N}_0$. Our bounds will involve various multiplicative constants $\mathrm{poly}(\cdot)$ which are polynomials in their input.

**SSL.** In this paper, we consider Semi-Supervised Learning (SSL) [8, 23] on an undirected graph of size $n$. We observe a weighted adjacency matrix $A = [a_{ij}]_{i,j=1}^n \in \mathbb{R}_+^{n \times n}$ as well as *node features* $z_1, \ldots z_n \in \mathbb{R}^p$ at each node of the graph. We also observe *some* labels $y_1, \ldots, y_{n_{\mathrm{tr}}} \in \mathbb{R}$ at training time and aim to predict the remaining labels $y_{n_{\mathrm{tr}}+1}, \ldots, y_n$. In a classification framework, $y \in \{-1, 1\}$. For simplicity, we assume that $n_{\mathrm{tr}}$ and $n_{\mathrm{te}} = n - n_{\mathrm{tr}}$ are both in $\mathcal{O}(n)$[1]. We denote by $Z \in \mathbb{R}^{n \times p}$ the matrix whose rows contain the node features, $Z_{\mathrm{tr}}, Z_{\mathrm{te}}$ respectively its first $n_{\mathrm{tr}}$ and last $n_{\mathrm{te}}$ rows, and similarly $Y_{\mathrm{tr}}, Y_{\mathrm{te}}$ the vectors containing the observed and non-observed labels.

**Graph smoothing with mean aggregation.** Here we consider a simplified situation of *linear* GNN with mean aggregation, that is, equation (2) with linear $\Psi$. Since all linear weights collapses into a

---

[1] while this is an important topic in SSL [4], here we do not focus on the number of needed labels and perform an asymptotic analysis instead.

single matrix, a linear GNN with $k$ layers just corresponds to performing $k$ rounds of mean aggregation on the node features, then learning on the smoothed features. We denote by $d_A = [\sum_i a_{ij}]_j \in \mathbb{R}^n_+$ the vector containing the degrees of the graph and $D = \mathrm{diag}(d_A)$. Assuming that all degrees are non-zero, performing one round of mean aggregation corresponds to multiplying $Z$ by $L = D^{-1}A$. Note that $\mathrm{Id} - L$ is then the *random walk Laplacian* of the graph. The smoothed node features after $k$ rounds of mean aggregation are:

$$Z^{(k)} = L^k Z\,.$$

Each row, denoted by $z_i^{(k)} \in \mathbb{R}^p$, contains the smoothed features of an individual node. Similar to the non-smoothed features, its first $n_{\text{tr}}$ and last $n_{\text{te}}$ rows are denoted $Z_{\text{tr}}^{(k)}, Z_{\text{te}}^{(k)}$.

**Learning.** In this paper, we consider learning with a Mean Square Error (MSE) loss and Ridge regularization. For $\lambda > 0$, the regression coefficients vector on the smoothed features is

$$\hat{\beta}^{(k)} \stackrel{\text{def.}}{=} \mathrm{argmin}_\beta \tfrac{1}{2n_{\text{tr}}} \left\| Y_{\text{tr}} - Z_{\text{tr}}^{(k)} \beta \right\|^2 + \lambda \left\| \beta \right\|^2 = \left( \tfrac{(Z_{\text{tr}}^{(k)})^\top Z_{\text{tr}}^{(k)}}{n_{\text{tr}}} + \lambda \mathrm{Id} \right)^{-1} \tfrac{(Z_{\text{tr}}^{(k)})^\top Y_{\text{tr}}}{n_{\text{tr}}} \qquad (3)$$

Then, the test risk is defined as

$$\mathcal{R}^{(k)} \stackrel{\text{def.}}{=} n_{\text{te}}^{-1} \left\| Y_{\text{te}} - \hat{Y}_{\text{te}}^{(k)} \right\|^2 \quad \text{where } \hat{Y}_{\text{te}}^{(k)} = Z_{\text{te}}^{(k)} \hat{\beta}^{(k)} \qquad (4)$$

It is well known that when $k \to \infty$, the matrix $L^k$ will converge to a matrix with constant rows, and $\mathcal{R}^{(\infty)} \stackrel{\text{def.}}{=} \lim_{k \to \infty} \mathcal{R}^{(k)}$ will just be close to the variance of $Y$, see Sec. 3 for a precise statement. Very often, this degrades the results with respect to doing a simple linear regression: $\mathcal{R}^{(0)} < \mathcal{R}^{(\infty)}$. Our goal is to illustrate some situations where a finite amount of smoothing provably improves the test risk, that is, there is an optimal $k^\star > 0$ such that $\mathcal{R}^{(k^\star)} < \min(\mathcal{R}^{(0)}, \mathcal{R}^{(\infty)})$.

**Random graph model.** To perform a fine-grained analysis of our problem, we need a statistical model linking the graph, the node features, and the labels. We adopt popular *latent space random graph models* akin to graphons [29]. Although such models are obviously idealized, we believe that they faithfully convey the main insights. In these models, to each node $i$ is associated an *unobserved latent variable* $x_i \in \mathbb{R}^d$ with $d \geqslant p$ (often $d \gg p$), and edge weights are assumed to be equal to $a_{ij} = W(x_i, x_j)$ where $W : \mathbb{R}^d \times \mathbb{R}^d \to \mathbb{R}_+$ is a *connectivity kernel*. Note that edges may also be taken as *random Bernoulli variables*, but we do not consider this here for simplicity. Moreover, we consider that the $(x_i, y_i)$ are drawn *iid* from some joint distribution, and the node features are a linear projection of the latent variables to a lower dimension: $z_i = M^\top x_i$ for some unknown $M \in \mathbb{R}^{d \times p}$ that satisfies $M^\top M = \mathrm{Id}_p$. At the end of the day:

$$\forall i, j, \quad (x_i, y_i) \stackrel{iid}{\sim} P, \quad z_i = M^\top x_i, \quad a_{ij} = W(x_i, x_j) \qquad (5)$$

For this model, note that

$$Z^{(k)} = L^k Z = L^k X M = X^{(k)} M \text{ where } X^{(k)} = L^k X$$

In other words, the smoothed node features $Z^{(k)}$ also correspond to a linear projection of the (unknown) *smoothed latent variables* $X^{(k)}$. To summarize, compared to "classical" machine learning on the $(x_i, y_i)$, we do *not* observe directly the $x_i$, but only a projection of them $z_i = M^\top x_i$. Although we assume that $M$ is orthogonal, we do *not* assume that it is "information-preserving" (e.g. it does not satisfy the Johnson-Lindenstrauss lemma), but rather that information *is* lost between the $x$ and the $z$. However, we also observe the graph $W(x_i, x_j)$. Our goal is illustrate how mean aggregation may restore some of the lost information.

In the rest of the paper, we use the Gaussian kernel with a small additive term $\varepsilon > 0$:

$$W(x, y) = \varepsilon + W_g(x, y) \quad \text{where } W_g(x, y) \stackrel{\text{def.}}{=} e^{-\frac{1}{2}\|x - y\|^2} \qquad (6)$$

The coefficient $\varepsilon$ is added to lower-bound the degrees of the graph and avoid degenerate situations. While this seems to be needed for our current proof technique, we use $\varepsilon = 0$ in Fig. 2 and 3. The Gaussian kernel is a classical model in theoretical graph machine learning [38].

# 3   Oversmoothing

In this section, we briefly examine the oversmoothing case, when $k \to \infty$ while all other parameters are fixed. In this case, it is well-known that all node features converge even for general GNNs [34]. For completeness, we state below this result in our settings. We have the following well-known ergodic theorem for stochastic matrices such as $L$.

**Theorem 1** (Ergodic theorem for stochastic matrices, e.g. [2, Thm. 4.2].). *Recall that $d_A$ is the vector of degrees, let $\bar{d} = d_A / d_A^\top 1_n$. We have*

$$L^k \xrightarrow[k \to \infty]{} 1_n \bar{d}^\top \tag{7}$$

This easily allows us to prove the next result.

**Corollary 1.** *We have the following*

$$\hat{Y}_{\text{te}}^{(k)} \xrightarrow[k \to \infty]{} \left( \frac{\|v\|^2}{\lambda + \|v\|^2} \bar{y}_{\text{tr}} \right) 1_{n_{\text{te}}} \tag{8}$$

*where $v = Z^\top \bar{d}$ and $\bar{y}_{\text{tr}} = n_{\text{tr}}^{-1} \sum_{i=1}^{n_{\text{tr}}} y_i$.*

*Proof.* We use Thm. 1 to get $L^k X M \to 1_n v^\top$, and $(\lambda \mathrm{Id} + vv^\top)^{-1} v = \frac{v}{\lambda + \|v\|^2}$. $\qquad \square$

Hence, in the limit $k \to \infty$, the predicted labels become all equal. When $\lambda \approx 0$, this value is, as expected, the average of the labels in the training set $\bar{y}_{\text{tr}}$. Using simple concentration inequalities, it is generally easy to show that $\mathcal{R}^{(\infty)} \approx \mathrm{Var}(y) + \mathcal{O}(1/\sqrt{n})$. In most cases, this leads to situations where $\mathcal{R}^{(0)} < \mathcal{R}^{(\infty)}$, that is, it is better to perform regression directly on the node features. In the next sections, we analyze some examples where smoothing provably helps.

# 4   Finite smoothing: Linear Regression

In this section, we consider a problem of linear regression on Gaussian data. We consider $x \sim \mathcal{N}_{0,\Sigma}$ for some positive definite covariance matrix $\Sigma$, and $y = x^\top \beta^\star$, without noise for simplicity (noise would just add an additional variance terms to all our bounds). We will first describe our main result that holds under a certain condition that is not necessarily easy to interpret, then give a sketch of proof in Sec. 4.1, and an example in dimension $d = 2$ where this assumption is satisfied in Sec. 4.2.

For a symmetric positive semi-definite matrix $S \in \mathbb{R}^{d \times d}$, we define the following function

$$R_{\text{reg.}}(S) \stackrel{\text{def.}}{=} (\Sigma^{\frac{1}{2}} \beta^\star)^\top \left( \mathrm{Id} - S^{\frac{1}{2}} M (\lambda \mathrm{Id} + M^\top S M)^{-1} M^\top S^{\frac{1}{2}} \right)^2 (\Sigma^{\frac{1}{2}} \beta^\star) \in \mathbb{R}_+ \tag{9}$$

where we recall that $M$ is the projection matrix to obtain the node features $z = M^\top x$. Note that it satisfies $0 \leqslant R(S) \leqslant \|\beta^\star\|_\Sigma^2$. Our result will be valid under the following assumption:

**Assumption 1.** *We have $R_{\text{reg.}}(\Sigma) > R_{\text{reg.}}((\mathrm{Id} + \Sigma^{-1})^{-2}\Sigma)$.*

Note that $(\mathrm{Id} + \Sigma^{-1})^{-2}\Sigma$ is indeed symmetric since $(\mathrm{Id} + \Sigma^{-1})^{-1}$ and $\Sigma$ permute. Our main result can be stated informally as follows, it is detailed in the next section along with a sketch of proof. Recall that the kernel is taken as (6).

**Theorem 2** (Existence of optimal smoothing for regression.). *Take any $\rho > 0$, and suppose that Assumption 1 holds. If $\varepsilon$ is sufficiently small and $n$ is sufficiently large, then with probability $1 - \rho$, there is $k^\star > 0$ such that $\mathcal{R}^{(k^\star)} < \min(\mathcal{R}^{(0)}, \mathcal{R}^{(\infty)})$.*

## 4.1   Sketch of proof

As we will see, it is easy to show that $\mathcal{R}^{(0)} < \mathcal{R}^{(\infty)}$ with high probability. Our main goal will therefore be to show that $\mathcal{R}^{(1)} < \mathcal{R}^{(0)}$ with high probability under Assumption 1, which is sufficient to show the existence of an optimal $k^\star \geqslant 1$. Using concentration inequalities, we will prove a rigorous non-asymptotic bound for $\mathcal{R}^{(1)}$. In the next section, we also derive an intuitive expression for $\mathcal{R}^{(k)}$ (although without rigorous proof), which we observe to match the numerics quite well.

The first step is to derive a closed form expression for $\mathcal{R}^{(0)}$, which is fairly easy using standard concentration techniques for subgaussian variables. The next result is proved in App. A.1.

**Theorem 3** (Regression risk without smoothing.). *With probability at least $1 - \rho$,*

$$\mathcal{R}^{(0)} = R_{\text{reg.}}(\Sigma) + \mathcal{O}\left(\frac{\|\Sigma\| \, \|\beta^\star\|^2 \, d\sqrt{\log(1/\rho)}}{(\lambda + \lambda_{\min})\sqrt{n}}\right) \tag{10}$$

*where $\lambda_{\min} = \lambda_{\min}(M^\top \Sigma M)$.*

As expected, when $p = d$, $M = \text{Id}$ and $\lambda \to 0$, we have $R_{\text{reg.}}(\Sigma) \to 0$ and the risk is exactly 0 in the infinite sample limit (recall that we have assumed zero noise on the labels). When $p < d$ however, the limit risk is generally non-zero. The worst case is obtained when $\Sigma\beta^\star$ is orthogonal to $M^\top$, where the risk reaches its maximum at $\|\beta^\star\|_\Sigma^2 = \mathbb{E}\,|y|^2$. Since this is the variance of $y$, this is also $\lim_{n\to\infty} \mathcal{R}^{(\infty)}$, hence we always have $\mathcal{R}^{(0)} \leqslant \mathcal{R}^{(\infty)}$ with high probability for $n$ large enough.

Let us now turn to computing the risk after one step of smoothing $k = 1$. We define $\Sigma^{(k)} = (\text{Id} + \Sigma^{-1})^{-2k}\Sigma$. The main result of this section is the following.

**Theorem 4** (Regression risk with one step of smoothing.). *With probability at least $1 - \rho$,*

$$\mathcal{R}^{(1)} = R_{\text{reg.}}(\Sigma^{(1)}) + \mathcal{O}\left(C\varepsilon^{1/5}\right) + \mathcal{O}\left(\frac{C'\log n\sqrt{d + \log(1/\rho)}}{(\lambda + \lambda_{\min})\sqrt{n}}\right) \tag{11}$$

*where $C = \text{poly}(\|\Sigma\|, e^d, |\text{Id} + \Sigma|)$, $C' = \text{poly}(\varepsilon^{-1}, \|\Sigma\|, \|\beta^\star\|)$ and $\lambda_{\min} = \lambda_{\min}(M^\top \Sigma^{(1)} M)$.*

This theorem gives a limiting expression or $\mathcal{R}^{(1)}$ with two additional error terms. The first goes to 0 with $\varepsilon$ and is due to the deviation from the kernel (6) to the exact Gaussian kernel $W_g$. The second term goes to 0 when $n \to \infty$ and is controlled via concentration inequalities. The limit risk when $\varepsilon \to 0$, $n \to \infty$ is $\mathcal{R}^{(1)} \approx R_{\text{reg.}}(\Sigma^{(1)})$, which is strictly lower than $R_{\text{reg.}}(\Sigma)$ by Assumption 1 and proves Theorem 3. Note that, to get $\mathcal{R}^{(1)} < \mathcal{R}^{(0)}$, we generally need $\varepsilon \lesssim e^{-d}$ and therefore $n \gtrsim e^d$, which seems to be an unavoidable artifact in our current proof technique.

Let us try to better understand Assumption 1 by sketching the proof of Thm. 4. The proof relies on an approximate description of the distribution of the smoothed node features $z_i^{(1)} = M^\top x_i^{(1)}$ where we recall that the $x_i^{(1)}$ are the rows of $X^{(k)} = L^k X$. We define $d(x) = |\text{Id} + \Sigma|^{-\frac{1}{2}} e^{-\frac{1}{2}\|x\|_{(\text{Id}+\Sigma)^{-1}}^2}$ and

$$\varphi_{\text{reg.}}(x) = \frac{d(x)}{d(x) + \varepsilon}(\Sigma^{-1} + \text{Id})^{-1}x\,. \tag{12}$$

Then, using some chaining concentration inequalities for subgaussian variables (Lemma 7 in the appendix) and properties of Gaussian distributions (Lemma 5), we can prove the following.

**Lemma 1.** *With probability at least $1 - \rho$, for all $i = 1, \ldots, n$:*

$$\left.\begin{array}{r}\left\|x_i^{(1)} - \varphi_{\text{reg.}}(x_i)\right\|_{\Sigma^{-1}} \\[2mm] \left\|\Sigma^{-\frac{1}{2}}\left(x_i^{(1)}(x_i^{(1)})^\top - \varphi_{\text{reg.}}(x_i)\varphi_{\text{reg.}}(x_i)^\top\right)\Sigma^{-\frac{1}{2}}\right\|\end{array}\right\} \lesssim \frac{C\log n(\sqrt{d + \log(1/\rho)})}{\sqrt{n}} \tag{13}$$

*where $C = \text{poly}(\varepsilon^{-1}, \|\Sigma\|, |\text{Id} + \Sigma|)$.*

Hence the smoothed latent variables behaves almost like $(\text{Id} + \Sigma^{-1})^{-1}x$, up to a deviation $\varepsilon$ that is handled in Lemma 3 in the appendix. The covariance of these data is $\Sigma^{(1)} = (\text{Id} + \Sigma^{-1})^{-2}\Sigma$, hence we can adapt the proof of Thm. 3 to obtain Thm. 4. All details are given in App. A.2.

### 4.2 Intuition and exact computation in dimension $d = 2$

We proved above that $x^{(1)}$ behaves almost like $(\text{Id} + \Sigma^{-1})^{-1}x$, whose covariance is $\Sigma^{(1)}$. Similarly, by applying repeated smoothing we can extrapolate that $x^{(k)}$ behaves like $(\text{Id} + \Sigma^{-1})^{-k}x$, such that $\mathcal{R}^{(k)} \approx R_{\text{reg.}}(\Sigma^{(k)})$. The rigorous proof of this fact becomes increasingly complicated and is skipped here. The matrix $\Sigma^{(k)}$ has the same eigendecomposition as $\Sigma$, but where every eigenvalue $\lambda_i$

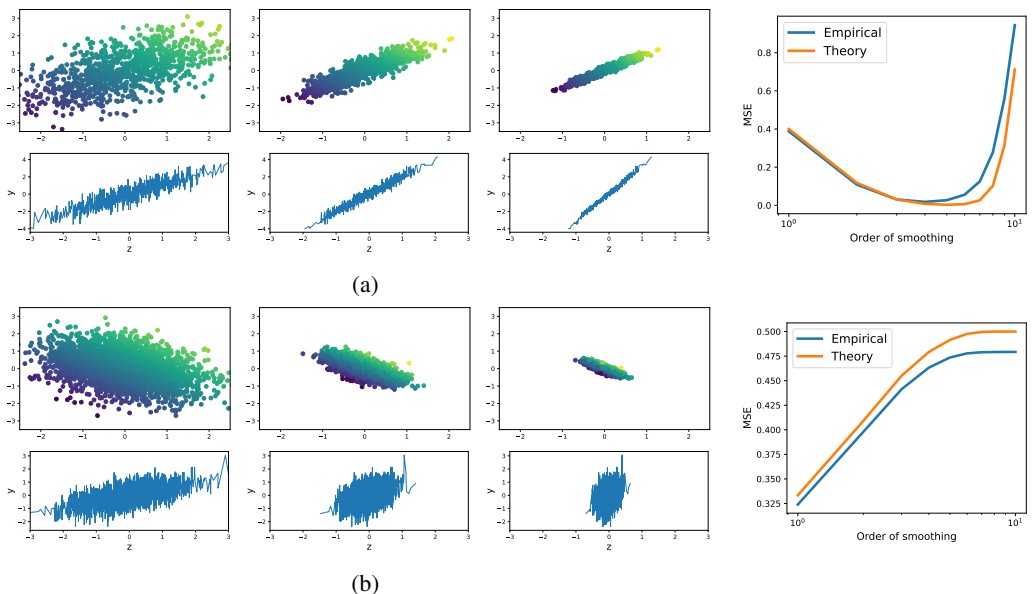

Figure 2: Illustration of mean aggregation smoothing on the regression example described in Sec. 4.2. **For both subfigures: First three figures on the left, top:** *unobserved* latent variables $X^{(k)}$ in dimension $d = 2$ where the colors are the $Y$; **bottom:** observed node features $Z^{(k)} = X^{(k)}M$ in dimension $p = 1$ on the x-axis, labels $Y$ on the y-axis. **From left to right**, three order of smoothing $k = 0, 1$ and $2$ are represented. **Figure on the right:** comparison of empirical and theoretical MSE given by (14) with respect to order of smoothing $k$. **Subfig. a:** $\lambda_1 = 2, \lambda_2 = 1/2$ (smoothing does help), **Subfig. b:** $\lambda_1 = 1/2, \lambda_2 = 1$ (smoothing does not help).

is replaced by $\lambda_i^{(k)} = (1 + 1/\lambda_i)^{-2k}\lambda_i$. This can be interpreted as follows: when $\lambda_i \gg 1$ is large, $\lambda_i^{(1)} \sim \lambda_i$, while if $\lambda_i \ll 1$ is small, $\lambda_i^{(1)} \sim \lambda_i^{2k+1}$ (note that the constant "1" here is due to our kernel (6), it is not inherently significant). Hence smoothing **shrinks the directions of the small eigenvalues faster than that of the large ones**. Thus, if $\beta^\star$ is mostly aligned with the eigenvectors of large eigenvalues, shrinking the small eigenvalues may *reduce unwanted noise* that emerges when projecting the node features $z = M^\top x$. On the other hand, if all eigenvalues of $\Sigma$ are equal, then $\Sigma^{(k)} \propto \Sigma$, and smoothing *does not help*, since in the limit $\lambda = 0$, the risk is invariant to scaling $R_{\text{reg.}}(aS) = R_{\text{reg.}}(S)$. Worse, we will see on an example below that smoothing can actually degrade the performance when $\beta^\star$ is unpropery aligned.

We illustrate this in dimension $d = 2$. Consider the following settings: $d = 2, p = 1$, $\Sigma$ has two eigenvalues $\lambda_1 \gg 1$ and $\lambda_2 \ll 1$, with respective eigenvectors $u_1 = [1, 1]/\sqrt{2}$ and $u_2 = [-1, 1]/\sqrt{2}$, and $\beta^\star$ is fully correlated with the first eigenvector: $\beta^\star = bu_1$. Finally, $M^\top = [1, 0]$ is the projection on the first coordinate. This situation is represented in Fig. 2. In this case, we can compute explicitly:

$$\mathcal{R}^{(k)} \approx R_{\text{reg.}}(\Sigma^{(k)}) = \lambda_1 b^2 \frac{(2\lambda + \lambda_2^{(k)})^2 + \lambda_2^{(k)}\lambda_1^{(k)}}{(2\lambda + \lambda_1^{(k)} + \lambda_2^{(k)})^2} \tag{14}$$

So, if $\lambda_2^{(k)}$ decreases faster than $\lambda_1^{(k)}$, this function will first decrease to a minimum of approximately $\lambda_1 b^2 \left(\frac{2\lambda}{2\lambda + \lambda_1^{(k^\star)}}\right)^2$ (when $\lambda_2^{(k)} \approx 0$), before increasing again to $\lambda_1 b^2 = \|\beta^\star\|_\Sigma^2 = \lim_{n \to \infty} \mathcal{R}^{(\infty)}$. This is illustrated in Fig. 2, for $\lambda_1 = 2$ and $\lambda_2 = 1/2$, where we empirically observes a minimum $k^\star$ that matches rather well the one predicted by (14).

**Homophily vs. Heterophily and a failure case**  In graph theory, *homophily* refers to the concept that linked nodes tend to display similar properties: for instance, friends on social networks have similar preferences, and so on. In graph machine learning, it generally means that linked nodes tend to have similar node features and labels. This concept is at the core of many graph signal processing and graph machine learning methods: for instance, spectral clustering is akin to a low-pass filter on the graph structure. However, it has been observed that real graphs may sometimes exhibit a low

level of homophily [51, 5]. They are rather said to be *heterophilic*, a somewhat less "well-defined" concept: in heterophilic graphs, linked nodes can be similar or dissimilar, some attributes can be homophilic and others heterophilic, and so on.

In our settings, at first glance it seems that our very regular random graph model always results in homophilic graphs, as the Gaussian kernel decreases with the distance between latent variables, and the latter are strongly linked with the node features. This is partly true, however is it also possible that nodes linked by a "strong" edge (with a high weight) have very different labels, which can be said to be a (toy) example of heterophily. For instance, consider the 2D linear regression example above given by (14). We have seen that when the regression vector is in the direction of the eigenvector corresponding to a high eigenvalue, then beneficial smoothing appears, as it reduces the noise in the observed node features (Fig. 2a). However, when the regression vector is instead in the low-eigenvalue direction, then close-by latent variables have very different labels, and the graph is more heterophilic. In this case, beneficial smoothing does *not* appear, and any smoothing strictly degrades the MSE! (Fig. 2b) This is due to the fact that in this case the "information" in node features vanishes faster than the noise. Of course, this is an exceedingly simple model of heterophily, and a better understanding and modelization of this phenomenon remains an outstanding open question.

**Discussion**  Recent literature on GNNs have adressed both oversmoothing and heterophily by clever normalization techniques [50, 17, 5, 37], combined with quantitative metrics of these phenomena [49, 51]. However, these tend to indiscriminately combat oversmoothing, without taking into account potential beneficial smoothing. In future work, our analysis could help designing more detailed normalization methods, e.g. after some estimation step that would identify which directions in the data are squeezed by smoothing, and which of them are relevant or not for learning.

## 5   Finite smoothing: classification

In this section, we examine a simple classification problem for two balanced classes with Gaussian distribution with identity covariance. The distribution of the labels and latent variables is:

$$(x, y) \sim (1/2)(\mathcal{N}_\mu \otimes \{1\} + \mathcal{N}_{-\mu} \otimes \{-1\}) \tag{15}$$

That is, with equal probability $x$ is drawn from $\mathcal{N}_\mu$ and $y = 1$, or $x \sim \mathcal{N}_{-\mu}$ and $y = -1$. As $\|\mu\|$ increases, the problem become simpler, there is an extensive literature on this problem [12, 40, 3]. Note that in this case $z_i$ are also Gaussian, with mean $\nu \overset{\text{def.}}{=} M^\top \mu$ or $-\nu$ and identity covariance.

We note that this is not a *difficult* problem *per se*, and that *linear regression with the MSE* is certainly not the method of choice to solve it: there are plethora of losses better adapted to binary classification such as the binary cross-entropy (left for future investigations), or even other dedicated methods: a Spectral Clustering algorithm on the graph alone would be able to perform the classification task under some mild hypotheses [40, 1] (without using the node features!). Nevertheless, let us recall that our main goal is to illustrate the smoothing phenomenon, and as we will see, the interpretation here will be quite different from the previous section. Our main result is the following.

**Theorem 5** (Existence of optimal smoothing for classification.)**.** *Take any $\rho > 0$. If $\varepsilon$ is sufficiently small, and $\|\mu\|, n$ are sufficiently large, and $\|M^\top \mu\| > 0$, then with probability $1 - \rho$, there is $k^\star > 0$ such that $\mathcal{R}^{(k^\star)} < \min(\mathcal{R}^{(0)}, \mathcal{R}^{(\infty)})$.*

Note that we have assumed $\|\mu\|$ to be sufficiently large here. However, we do *not* assume that $\|M^\top \mu\|$ is large (just non-zero), and the classification problem on the $z_i$ alone may be very difficult. The rest of this section presents a sketch of proof and intuitions behind this theorem.

### 5.1   Sketch of proof and intuition

As in the previous section, it will be easy to show that $\mathcal{R}^{(0)} < \mathcal{R}^{(\infty)}$ with high probability, and we will prove that $\mathcal{R}^{(1)} < \mathcal{R}^{(0)}$ with high probability. Again, we start by providing an expression for $\mathcal{R}^{(0)}$. For $s \in \mathbb{R}_+$, we define the following function

$$R_{\text{cl.}}(s) = \frac{(s + \lambda)^2 + s \|\nu\|^2}{(s + \lambda + \|\nu\|^2)^2} \tag{16}$$

The next result is proved in App. B.1. Recall that $\nu = M^\top \mu$.

**Theorem 6** (Classification risk without smoothing.)**.** *With probability at least $1 - \rho$,*

$$\mathcal{R}^{(0)} = R_{\text{cl.}}(1) + \mathcal{O}\left(\frac{\|\nu\|^4 \, p\sqrt{\log(1/\rho)}}{\sqrt{n}}\right) \tag{17}$$

When $\|\nu\| \to \infty$, the risk goes to 0, as expected, since the Gaussians get further and further away. However, when $\|\nu\| \to 0$, which can happen *either* when $\|\mu\|$ is small or when $M$ becomes orthogonal to $\mu$, the risk goes to 1, its worst value, for random guesses. Since it is also the variance of $y$, we have indeed $\mathcal{R}^{(0)} \leqslant 1 \approx \mathcal{R}^{(\infty)}$ with high probability for $n$ large enough.

Let us now turn to computing the risk after one step of smoothing $k = 1$. The main result of this section is the following.

**Theorem 7** (Classification risk with one step of smoothing.)**.** *With probability at least $1 - \rho$,*

$$\mathcal{R}^{(1)} = R_{\text{cl.}}(1/4) + \mathcal{O}\left(C\left(\varepsilon^{\frac{1}{4}} + \frac{1}{\varepsilon^3}e^{-\frac{\|\mu\|^2}{4}}\right)\right) + \mathcal{O}\left(\frac{C'(\log n)(\sqrt{d + \log(1/\rho)})}{\sqrt{n}}\right) \tag{18}$$

*where $C = \text{poly}(\|\mu\|, e^d)$ and $C' = \text{poly}(\varepsilon^{-1}, \|\mu\|)$.*

This theorem show that $\mathcal{R}^{(1)} \approx R_{\text{cl.}}(1/4)$ with two additional error terms. First of all, a quick function study shows that $R_{\text{cl.}}(1/4) < R_{\text{cl.}}(1)$ when $\|\nu\| > 0$, which shows Thm. 5 when the errors are small enough. The last error term goes to 0 when $n \to \infty$ and is controlled via concentration inequalities. The first one is small when $\varepsilon$ is small and $\|\mu\|$ is large enough. We remark that, unlike the previous section where the error terms vanished in the limit $\varepsilon \to 0, n \to \infty$, here there is a non-zero error term due to $\|\mu\|$ whose explicit expression is still open. Hence, for instance, the discrepancy between the empirical observations and the theory in Fig. 3 compared to Fig. 2. Note that, as in the previous section, we need at least $\varepsilon \lesssim e^{-d}$ and $n \gtrsim e^d$. However, here we also need $\|\mu\| \gtrsim \sqrt{d}$. This rate is similar to early analyses of Gaussian Mixture learning [12], although they have been greatly improved since [3].

As previously, we define here $d_\mu(x) \overset{\text{def.}}{=} 2^{-d/2}e^{-\frac{\|x-\mu\|^2}{4}}$, and

$$\varphi_{\text{cl.}}(x) = \frac{d_\mu(x)\left(\frac{x+\mu}{2}\right) + d_{-\mu}(x)\left(\frac{x-\mu}{2}\right)}{2\varepsilon + d_\mu(x) + d_{-\mu}(x)} \tag{19}$$

The following result is similar to Lemma 1 and is shown in App. B.2.

**Lemma 2.** *With probability at least $1 - \rho$,*

$$\left.\begin{array}{c}\sup\limits_{i=1,\ldots,n}\left\|x_i^{(1)} - \varphi_{\text{cl.}}(x_i)\right\| \\ \sup\limits_{i=1,\ldots,n}\left\|x_i^{(1)}(x_i^{(1)})^\top - \varphi_{\text{cl.}}(x_i)\varphi_{\text{cl.}}(x_i)^\top\right\|\end{array}\right\} \lesssim \frac{\text{poly}(\varepsilon^{-1})\log n(\sqrt{d} + \sqrt{\log(1/\rho)})}{\sqrt{n}} \tag{20}$$

Let us now examine $\varphi_{\text{cl.}}(x)$ closer. In the limit $\varepsilon \to 0$, $\varphi_{\text{cl.}}(x)$ is a convex combination of $(x + \mu)/2$ and $(x - \mu)/2$. Hence, when $x \sim \mathcal{N}_\mu$, with high probability $x$ is close to $\mu$ and $d_\mu(x) \gg d_{-\mu}(x)$, and in this case, $\varphi_{\text{cl.}}(x) \approx \frac{x+\mu}{2}$, whose distribution is $\mathcal{N}_{\mu,\text{Id}/4}$. The same reasoning applies to the other community. Hence, up to some error $\mathcal{O}\left(e^{-\|\mu\|^2/4}\right)$ due to the communities getting closer to each other, **the smoothed features in each community have the same mean but a reduced variance** $\text{Id}/4$, thus the limit risk $R_{\text{cl.}}(1/4)$ in our limit expression for $\mathcal{R}^{(1)}$. In other words, **the communities shrink faster than they collapses together**, and this reflects on the projected node features. An illustration of this phenomenon is given in Fig. 3. All proof details are in App. B.2.

## 5.2 Numerical illustration

In light of the proof of the theorem above, when $x_i$ belongs to the first community and $x_i^{(1)} \approx \varphi_{\text{cl.}}(x_i) \approx \frac{x_i+\mu}{2}$, applying a second smoothing would transform it to $\frac{\varphi_{\text{cl.}}(x_i)+\mu}{2} \approx \frac{x_i+3\mu}{4}$, that

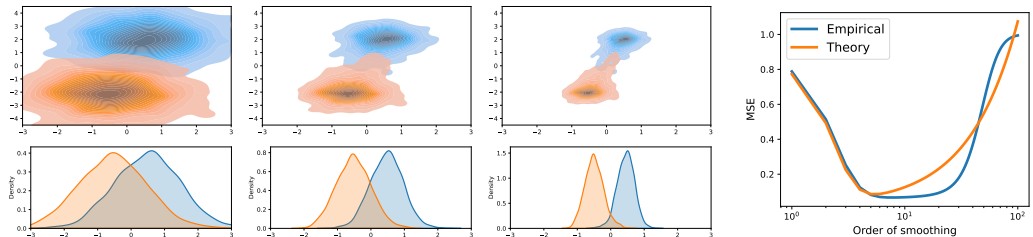

Figure 3: Illustration of mean aggregation smoothing on a classification task with two Gaussians with dimensions $d = 2$, $p = 1$, where $M$ projects on the first coordinate. **First three figures on the left, top:** density of *unobserved* latent variables $X^{(k)}$ in dimension $d = 2$; **bottom:** density of observed node features $Z^{(k)} = X^{(k)}M$ in dimension $p = 1$. **From left to right**, three order of smoothing $k = 0, 1$ and 2 are represented (recall that the smoothing is agnostic to the labels, we cannot perform in-community smoothing). **Figure on the right:** comparison of empirical and theoretical MSE given by (21) with respect to order of smoothing $k$. For low $k$, the node features communities are indeed more and more separated, and learning improves.

is, it keeps the same mean but now has variance $\mathrm{Id}/16$. If we look at the proof, the error term $\mathcal{O}\left(e^{-\|\mu\|^2/4}\right)$ would become in this case $e^{-\frac{\|\mu\|^2}{2(1+1/4)}}$. While this expression is far from being exact and we do not a rigorous proof here (which seems far more complex than the case $k = 1$), we can infer some approximate expression:

$$\mathcal{R}^{(k)} \approx R_{\mathrm{cl.}}(4^{-k}) + \mathcal{O}\left(\sum_{\ell=0}^{k-1} e^{-\frac{\|\mu\|^2}{2(1+4^{-\ell})}}\right) \tag{21}$$

Unlike the expression (14), the term $R_{\mathrm{cl.}}(4^{-k})$ is strictly decreasing when $k$ increases. Oversmoothing is modelled by the error term, for which we do not have an exact expression, and for which we suspect that the quality of approximation degrades as $k$ increases. Nevertheless, we evaluate this expression on an example in Fig. 3 (with an adjusted multiplicative constant for the error term in (21)) and find that it is a reasonably good approximation, at least for small $k$.

## 6    Conclusion and outlooks

While the oversmoothing phenomenon $k \to \infty$ has been well characterized, until now there has been no theoretical studies that rigorously modelled both the benefits of finite smoothing before oversmoothing kicks in. In this paper, we adopted a simplified context of linear GNNs with mean aggregation and random graphs with partially observed latent variables, and proved on two representative examples the co-existence of both phenomena. We identified two mechanisms for the benefits of mean aggregation: it tends to shrink noisy principal components faster than meaningful ones, and it tends to gather nodes of the same community faster than they collapses together. We obtained theoretical expressions up to some error terms that matched the numerics quite well on simple synthetic data.

There are many outlooks to this work. First and foremost, deriving inspiration from our theoretical observations to design better methods of setting the order of smoothing in practical application is a major challenge. As seen in Fig. 1 in the introduction, both mechanisms that we identified seem to come into play on real data. However, many quantities appearing in the risks (14) and (21) need to be estimated. Second, extending our theory to more complex loss functions (especially for classification) and non-linear GNNs is crucial. Finally, our work is a step towards a better understanding of *the relationship between node features and graph structure*, which is at the heart of (over)smoothing, heterophily, and all graph machine learning methods. A more general theory, and more realistic models of random graphs to analyze it, is still an open question.

#### Acknowledgment

This work was supported by ANR JCJC GRandMa (ANR-21-CE23-0006). NK thanks S. Vaiter for inspiring discussions.

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
