## Appendix

This appendix contains all proof of our results. We start by Linear Regression in App. A, then classification in App. B. App. C contains technical Lemmas. At their core, the proofs are a combination of chaining concentration inequalities for subgaussian variables and derivations on Gaussian distributions.

## A Linear Regression

### A.1 Proof of Theorem 3

*Proof.* Let us begin by the concentration of the optimal $\hat{\beta} = (\lambda \mathrm{Id} + Z_{\mathrm{tr}}^\top Z_{\mathrm{tr}}/n)^{-1} Z_{\mathrm{tr}}^\top Y_{\mathrm{tr}}/n$.

$$\frac{1}{n_{\mathrm{tr}}} Z_{\mathrm{tr}}^\top Z_{\mathrm{tr}} = \frac{1}{n_{\mathrm{tr}}} \sum_{i=1}^{n_{\mathrm{tr}}} z_i z_i^\top$$

By an application of [41, Corollary 5.50], which is a concentration inequality for covariance estimates of subgaussian vectors, we get that with probability at least $1 - \rho$,

$$\left\| \frac{1}{n_{\mathrm{tr}}} \sum_i z_i z_i^\top - M^\top \Sigma M \right\| \lesssim \frac{p\sqrt{\log(1/\rho)}}{\sqrt{n}}$$

since $n_{\mathrm{tr}} = \mathcal{O}(n)$. In particular, for $n$ large enough, we get $\lambda_{\min}(\frac{1}{n_{\mathrm{tr}}} \sum_i z_i z_i^\top) \geqslant \lambda_{\min}(M^\top \Sigma M)/2$. Similarly,

$$\frac{1}{n_{\mathrm{tr}}} Z_{\mathrm{tr}}^\top Y_{\mathrm{tr}} = \frac{M^\top}{n_{\mathrm{tr}}} X_{\mathrm{tr}}^\top X_{\mathrm{tr}} \beta^\star$$

and using the same concentration on the covariance of $X$ we obtain

$$\left\| \frac{1}{n_{\mathrm{tr}}} Z_{\mathrm{tr}}^\top Y_{\mathrm{tr}} - M^\top \Sigma \beta^\star \right\| \lesssim \frac{\|\beta^\star\| p\sqrt{\log(1/\rho)}}{\sqrt{n}}$$

At the end of the day

$$\left\| \hat{\beta} - (\lambda \mathrm{Id} + M^\top \Sigma M)^{-1} M^\top \Sigma \beta^\star \right\| \lesssim \frac{\|\beta^\star\| p\sqrt{\log(1/\rho)}}{(\lambda + \lambda_{\min}(M^\top \Sigma M))\sqrt{n}}$$

Let us now compute the limit of the test risk:

$$\frac{1}{n_{\mathrm{te}}} \left\| Y_{\mathrm{te}} - Z_{\mathrm{te}}^\top \hat{\beta} \right\|^2 = (\beta^\star)^\top \frac{1}{n_{\mathrm{te}}} X_{\mathrm{te}}^\top X_{\mathrm{te}} \beta^\star - 2\hat{\beta}^\top \left( \frac{1}{n_{\mathrm{te}}} Z_{\mathrm{te}}^\top Y_{\mathrm{te}} \right) + \hat{\beta}^\top \left( \frac{1}{n_{\mathrm{te}}} Z_{\mathrm{te}}^\top Z_{\mathrm{te}} \right) \hat{\beta}^\top$$

By a reasoning identical to the one above on $\frac{1}{n_{\mathrm{te}}} X_{\mathrm{te}}^\top X_{\mathrm{te}} \approx \Sigma$, $\frac{1}{n_{\mathrm{te}}} Z_{\mathrm{te}}^\top Y_{\mathrm{te}} \approx M^\top \Sigma \beta^\star$ and $\frac{1}{n_{\mathrm{te}}} Z_{\mathrm{te}}^\top Z_{\mathrm{te}} \approx M^\top \Sigma M$, and using $\left\| (\lambda \mathrm{Id} + M^\top \Sigma M)^{-1} M^\top \Sigma \beta^\star \right\| \leqslant \|\beta^\star\|$ (which can be seen using an SVD decomposition of $M\Sigma^{\frac{1}{2}}$) and $\|M\| \leqslant \|1\|$, after some computation we obtain

$$\mathcal{R}^{(0)} = (\Sigma^{\frac{1}{2}} \beta^\star)^\top \left( \mathrm{Id} - \Sigma^{\frac{1}{2}} M^\top (\lambda \mathrm{Id} + M^\top \Sigma M)^{-1} M\Sigma^{\frac{1}{2}} \right)^2 \Sigma^{\frac{1}{2}} \beta^\star$$

$$+ \mathcal{O}\left( \frac{\|\Sigma\| \|\beta^\star\|^2 d\sqrt{\log(1/\rho)}}{(\lambda + \lambda_{\min}(M^\top \Sigma M))\sqrt{n}} \right)$$

The first term is $R_{\mathrm{reg.}}(\Sigma)$, we use a union bound over all these inequalities to conclude the proof. $\square$

### A.2 Proof of Theorem 4

We start with the proof of Lemma 1. A small reminder on subgaussian variables is given in App. C.

*Proof of Lemma 1.* The proof relies on chaining concentration inequalities for subgaussian variables and properties of Gaussian distributions (Lemma 5 and 7 in App. C).

Note that $\|\varphi_{\text{reg.}}(x)\|_{\Sigma^{-1}} \leqslant \frac{\left\|\Sigma^{\frac{1}{2}}(\text{Id}+\Sigma)^{-\frac{1}{2}}\right\|}{\varepsilon} d(x) \|x\|_{(\text{Id}+\Sigma)^{-1}} \lesssim \frac{1}{\varepsilon}$. Moreover, by Lemma 5, $\mathbb{E}W_g(x, X) = d(x)$ and $\mathbb{E}W_g(x, X)X = d(x)(\Sigma^{-1} + \text{Id})^{-1}x$.

We decompose

$$
\begin{aligned}
\left\|x_i^{(1)} - \varphi_{\text{reg.}}(x_i)\right\|_{\Sigma^{-1}} &= \left\|\frac{\varepsilon \sum_j x_j + \sum_j W_g(x_i, x_j)x_j}{n\varepsilon + \sum_j W_g(x_i, x_j)} - \varphi_{\text{reg.}}(x_i)\right\|_{\Sigma^{-1}} \\
&= \left\|\frac{\varepsilon \frac{1}{n} \sum_j x_j + \frac{1}{n} \sum_j W_g(x_i, x_j)x_j}{\varepsilon + \frac{1}{n} \sum_j W_g(x_i, x_j)} - \frac{d(x)(\Sigma^{-1} + \text{Id})^{-1}x}{d(x) + \varepsilon}\right\|_{\Sigma^{-1}} \\
&\leqslant \frac{1}{n}\left\|\sum_j x_j\right\|_{\Sigma^{-1}} + \left\|\frac{\frac{1}{n} \sum_j W_g(x_i, x_j)x_j - d(x)(\Sigma^{-1} + \text{Id})^{-1}x}{\varepsilon + \frac{1}{n} \sum_j W_g(x_i, x_j)}\right\|_{\Sigma^{-1}} \\
&\quad + \left\|d(x)(\Sigma^{-1} + \text{Id})^{-1}x\right\|_{\Sigma^{-1}}\left|\frac{1}{\varepsilon + \frac{1}{n} \sum_j W_g(x_i, x_j)} - \frac{1}{\varepsilon + d(x)}\right| \\
&\leqslant \frac{1}{n}\left\|\sum_j x_j\right\|_{\Sigma^{-1}} + \frac{1}{\varepsilon}\left\|\frac{1}{n} \sum_j W_g(x_i, x_j)x_j - d(x)(\Sigma^{-1} + \text{Id})^{-1}x\right\|_{\Sigma^{-1}} \\
&\quad + \frac{1}{\varepsilon^2}\left|\frac{1}{n} \sum_j W_g(x_i, x_j) - d(x)\right|
\end{aligned}
$$

For the first term, applying Lemma 7 with $W = 1$, with probability $1 - \rho$ we have

$$
\frac{1}{n}\left\|\sum_j x_j\right\| \lesssim \frac{\log n \left(\sqrt{d} + \sqrt{\log(1/\rho)}\right)}{\sqrt{n}}
$$

Similarly, since $W_g$ is $C_L$ Lipschitz in the first variable with respect to $\|\cdot\|_{\Sigma^{-1}}$ with $C_L \lesssim \left\|\Sigma^{\frac{1}{2}}\right\|$, applying Lemma 7 we get

$$
\left\|\frac{1}{n} \sum_j W_g(x_i, x_j)x_j - d(x)(\Sigma^{-1} + \text{Id})^{-1}x\right\|_{\Sigma^{-1}} \lesssim \frac{\log n \left\|\Sigma^{\frac{1}{2}}\right\| \left(\sqrt{d} + \sqrt{\log(1/\rho)}\right)}{\sqrt{n}}
$$

and

$$
\left|\frac{1}{n} \sum_j W_g(x_i, x_j) - d(x)\right| \lesssim \frac{\sqrt{\log n} \left\|\Sigma^{\frac{1}{2}}\right\| \left(\sqrt{d} + \sqrt{\log(1/\rho)}\right)}{\sqrt{n}}
$$

which concludes the proof of the first inequality. The second is obtained by decomposing and using $\|\varphi_{\text{reg.}}(x)\|_{\Sigma^{-1}} \lesssim 1/\varepsilon$. $\qquad\square$

We will also need the following Lemma, to bound the deviation brought by $\varepsilon$ in the expression for $\varphi_{\text{reg.}}$.

**Lemma 3.** *We have*

$$
\left.\begin{aligned}
&\left\|\mathbb{E}\varphi_{\text{reg.}}(x)x^\top - (\Sigma^{(1)})^{\frac{1}{2}}\Sigma^{\frac{1}{2}}\right\| \\
&\left\|\mathbb{E}\varphi(x)\varphi_{\text{reg.}}(x)^\top - \Sigma^{(1)}\right\|
\end{aligned}\right\} \lesssim \text{poly}(e^d, |\text{Id} + \Sigma|)\varepsilon^{1/5}
$$

*Proof.* Denote by $\mathcal{B}_r = \{x; \|x\|_{\Sigma^{-1}} \leqslant r\}$. Within this ball, since $\|x\|_{(\mathrm{Id}+\Sigma)^{-1}} \leqslant \left\|(\mathrm{Id}+\Sigma)^{-\frac{1}{2}}\Sigma^{\frac{1}{2}}\right\|\|x\|_{\Sigma^{-1}} \leqslant \|x\|_{\Sigma^{-1}}$, we have $1/d(x) \leqslant |\mathrm{Id}+\Sigma|^{\frac{1}{2}} e^{r^2/2}$. We also recall that $\int \|x\|_{\Sigma^{-1}}^2 \mathcal{N}_{0,\Sigma}(x)dx \lesssim d$ and $\int_{\mathcal{B}_r^c} \|x\|_{\Sigma^{-1}}^2 \mathcal{N}_{0,\Sigma}(x)dx \lesssim 2^{d/2}e^{-r^2/4}$. Now we decompose, using $\left\|(\mathrm{Id}+\Sigma^{-1})^{-1}x\right\|_{\Sigma^{-1}} \leqslant \left\|\Sigma^{-\frac{1}{2}}(\mathrm{Id}+\Sigma^{-1})^{-1}\Sigma^{\frac{1}{2}}\right\|\|x\|_{\Sigma^{-1}}$,

$$
\begin{aligned}
\mathbb{E}\left\|\varphi_{\mathrm{reg.}}(x) - (\mathrm{Id}+\Sigma^{-1})^{-1}x\right\|_{\Sigma^{-1}}^2 &= \int \left\|\frac{\varepsilon}{d(x)+\varepsilon}(\mathrm{Id}+\Sigma^{-1})^{-1}x\right\|_{\Sigma^{-1}}^2 \mathcal{N}_{0,\Sigma}(x)dx \\
&\lesssim \int_{\mathcal{B}_r} \left\|\frac{\varepsilon}{d(x)+\varepsilon}(\mathrm{Id}+\Sigma^{-1})^{-1}x\right\|_{\Sigma^{-1}}^2 \mathcal{N}_{0,\Sigma}(x)dx \\
&\quad + \int_{\mathcal{B}_r^c} \left\|\frac{\varepsilon}{d(x)+\varepsilon}(\mathrm{Id}+\Sigma^{-1})^{-1}x\right\|_{\Sigma^{-1}}^2 \mathcal{N}_{0,\Sigma}(x)dx \\
&\lesssim \varepsilon^2 |\mathrm{Id}+\Sigma| e^{r^2} d + 2^{d/2}e^{-r^2/4} \\
&\lesssim d2^{d/2} |\mathrm{Id}+\Sigma| \varepsilon^{2/5}
\end{aligned}
$$

Where the last line is obtained by choosing $r = \sqrt{(8/5)\log(1/\varepsilon)}$. Then we use that for two random variables $X$ and $Y$, $\|\mathbb{E}X - \mathbb{E}Y\| \leqslant \sqrt{\mathbb{E}\|X-Y\|^2}$, and $\left\|\mathbb{E}XY^\top - \mathbb{E}YY^\top\right\| \leqslant \sqrt{\mathbb{E}\|Y\|^2}\sqrt{\mathbb{E}\|X-Y\|^2}$, and $\left\|\mathbb{E}XX^\top - \mathbb{E}YY^\top\right\| \leqslant (\sqrt{\mathbb{E}\|X\|^2} + \sqrt{\mathbb{E}\|Y\|^2})\sqrt{\mathbb{E}\|X-Y\|^2}$, to conclude. $\qquad\square$

We are now ready to show Theorem 4

*Proof of Theorem 4.* We proceed in two steps. First, we use Lemma 1 to show that we can replace the $x^{(1)}$ by $\varphi_{\mathrm{reg.}}(x_i)$ in the computation of the risk. Second, we use Lemma 3 to concentrate the $\varphi_{\mathrm{reg.}}(x_i)$ around their new expectations. We define $\hat{\beta}^\varphi$ and $\mathcal{R}^\varphi$ by replacing $Z^{(k)}$ with $Z^\varphi = X^\varphi M$ where the rows of $X^\varphi$ are the $\varphi_{\mathrm{reg.}}(x_i)$, in (3) and (4).

Since $\varphi_{\mathrm{reg.}}(x)$ is bounded, it is a subgaussian vector. We can therefore apply the same reasoning as in the proof of Theorem 6 and concentrate $(Z_{\mathrm{tr}}^\varphi)^\top Z_{\mathrm{tr}}^\varphi / n_{\mathrm{tr}}$. Using Lemma 4, for $\varepsilon$ small enough, and $n$ large enough, it is almost equal to $M^\top \Sigma^{(1)} M$, and thus $\lambda_{\min}((Z_{\mathrm{tr}}^\varphi)^\top Z_{\mathrm{tr}}^\varphi / n_{\mathrm{tr}}) \gtrsim \lambda_{\min}(M^\top \Sigma^{(1)} M)$. Finally, using Lemma 2, for $n$ large enough $\lambda_{\min}((Z_{\mathrm{tr}}^{(1)})^\top Z_{\mathrm{tr}}^{(1)} / n_{\mathrm{tr}}) \geqslant \lambda_{\min}((Z_{\mathrm{tr}}^\varphi)^\top Z_{\mathrm{tr}}^\varphi / n_{\mathrm{tr}})/2$. Since $\|\varphi_{\mathrm{reg.}}(x)\|_{\Sigma^{-1}} \lesssim 1/\varepsilon$, we bound

$$
\begin{aligned}
\left\|\hat{\beta} - \hat{\beta}^\varphi\right\| &= \left\|(\lambda\mathrm{Id} + (Z_{\mathrm{tr}}^{(1)})^\top Z_{\mathrm{tr}}^{(1)}/n)^{-1}(Z_{\mathrm{tr}}^{(1)})^\top Y_{\mathrm{tr}}/n_{\mathrm{tr}} - (\lambda\mathrm{Id} + (Z_{\mathrm{tr}}^\varphi)^\top Z_{\mathrm{tr}}^\varphi/n)^{-1}(Z_{\mathrm{tr}}^\varphi)^\top Y_{\mathrm{tr}}/n_{\mathrm{tr}}\right\| \\
&\leqslant \left\|((\lambda\mathrm{Id} + (Z_{\mathrm{tr}}^{(1)})^\top Z_{\mathrm{tr}}^{(1)}/n_{\mathrm{tr}})^{-1} - (\lambda\mathrm{Id} + (Z_{\mathrm{tr}}^\varphi)^\top Z_{\mathrm{tr}}^\varphi/n_{\mathrm{tr}})^{-1})(Z_{\mathrm{tr}}^\varphi)^\top Y_{\mathrm{tr}}/n_{\mathrm{tr}}\right\| \\
&\quad + \left\|(\lambda\mathrm{Id} + (Z_{\mathrm{tr}}^{(1)})^\top Z_{\mathrm{tr}}^{(1)}/n_{\mathrm{tr}})^{-1}((Z_{\mathrm{tr}}^{(1)})^\top Y_{\mathrm{tr}}/n_{\mathrm{tr}} - (Z_{\mathrm{tr}}^\varphi)^\top Y_{\mathrm{tr}}/n_{\mathrm{tr}})\right\| \\
&\leqslant \frac{\left\|M^\top \Sigma^{\frac{1}{2}}\right\|^2}{\varepsilon(\lambda + \lambda_{\min}(M^\top \Sigma^{(1)} M))^2} \sup_i \left\|\Sigma^{-\frac{1}{2}}\left(x_i^{(1)}(x_i^{(1)})^\top - \varphi(x_i)\varphi(x_i)^\top\right)\Sigma^{-\frac{1}{2}}\right\| \\
&\quad + \frac{\left\|M^\top \Sigma^{\frac{1}{2}}\right\|}{\lambda + \lambda_{\min}(M^\top \Sigma^{(1)} M)} \sup_i \left\|x_i^{(1)} - \varphi(x_i)\right\|_{\Sigma^{-1}} \\
&\lesssim \frac{\mathrm{poly}(\|\Sigma\|, \varepsilon^{-1})(\sqrt{d} + \sqrt{\log(1/\rho)})}{(\lambda + \lambda_{\min}(M^\top \Sigma^{(1)} M))^2\sqrt{n}}
\end{aligned}
$$

Using the same bounds on $\frac{1}{n_{\mathrm{te}}}(Z_{\mathrm{te}}^{(1)})^\top Y_{\mathrm{te}}$ and $\frac{1}{n_{\mathrm{te}}}(Z_{\mathrm{te}}^{(1)})^\top Z_{\mathrm{te}}^{(1)}$, we get

$$
\left|\mathcal{R}^{(1)} - \mathcal{R}^\varphi\right| \lesssim \frac{\mathrm{poly}(\|\Sigma\|, \varepsilon^{-1})(\sqrt{d} + \sqrt{\log(1/\rho)})}{(\lambda + \lambda_{\min}(M^\top \Sigma^{(1)} M))^2\sqrt{n}} \tag{22}
$$

Finally, we apply the same reasoning as in the proof of Theorem 3, we obtain

$$\mathcal{R}^{\varphi} = \|\beta^{\star}\|_{\Sigma}^2 - 2(M^{\top}\Sigma_{\varphi,x}\beta^{\star})^{\top}(\lambda\mathrm{Id} + M^{\top}\Sigma_{\varphi}M)^{-1}M^{\top}\Sigma_{\varphi,x}\beta^{\star}$$
$$+ (M^{\top}\Sigma_{\varphi,x}\beta^{\star})^{\top}(\lambda\mathrm{Id} + M^{\top}\Sigma_{\varphi}M)^{-1}\Sigma_{\varphi}(\lambda\mathrm{Id} + M^{\top}\Sigma_{\varphi}M)^{-1}M^{\top}\Sigma_{\varphi,x}\beta^{\star}$$
$$+ \mathrm{poly}(\varepsilon^{-1}, \|\Sigma\|, \|\beta^{\star}\|)\frac{\log n(\sqrt{d} + \sqrt{\log(1/\rho)})}{(\lambda + \lambda_{\min}(M^{\top}\Sigma^{(1)}M))\sqrt{n}}$$

where $\Sigma_{\varphi} = \mathbb{E}\varphi(x)\varphi(x)^{\top}$ and $\Sigma_{\varphi,x} = \mathbb{E}\varphi(x)x^{\top}$. We use Lemma 3 and a union bound over all these inequalities to conclude the proof. $\qquad\square$

# B  Classification of Gaussian mixtures

## B.1  Proof of Theorem 6

*Proof.* The proof is similar to that of Theorem 3. We begin by the concentration of the optimal $\hat{\beta} = (\lambda\mathrm{Id} + Z_{\mathrm{tr}}^{\top}Z_{\mathrm{tr}}/n_{\mathrm{tr}})^{-1}Z_{\mathrm{tr}}^{\top}Y_{\mathrm{tr}}/n_{\mathrm{tr}}$. We denote by $I_1, I_{-1} \subset \{1, \ldots, n_{\mathrm{tr}}\}$ the indices of the $x_i$ respectively from the first and second community, of size $n_1$ and $n_{-1}$. We have

$$\frac{1}{n_{\mathrm{tr}}}Z_{\mathrm{tr}}^{\top}Z_{\mathrm{tr}} = \frac{1}{n_{\mathrm{tr}}}\sum_i z_i z_i^{\top}$$
$$= \frac{n_1}{n_{\mathrm{tr}}}\frac{1}{n_1}\sum_{I_1} z_i z_i^{\top} + \frac{n_{-1}}{n}\frac{1}{n_{-1}}\sum_{I_{-1}} z_i z_i^{\top}$$

Since the communities are balanced, by a simple application of Hoeffding's inequality, with probability at least $1 - \rho$ we have $n_1/n_{\mathrm{tr}} = 1/2 + \mathcal{O}\left(\sqrt{\log(1/\rho)/n}\right)$. Then, as in the proof of Theorem 3 by an application of [41, Corollary 5.50], with probability at least $1 - \rho$,

$$\left\|\frac{1}{n_1}\sum_{I_1} z_i z_i^{\top} - \mathbb{E}_{\mathcal{N}_{\nu}}zz^{\top}\right\| \lesssim \frac{p\sqrt{\log(1/\rho)}}{\sqrt{n}}$$

and $\mathbb{E}_{\mathcal{N}_{\mu}}zz^{\top} = \nu\nu^{\top} + \mathrm{Id}$. We apply the same reasoning for $I_{-1}$, and we obtain

$$\left\|\frac{1}{n_{\mathrm{tr}}}Z_{\mathrm{tr}}^{\top}Z_{\mathrm{tr}} - (\mathrm{Id} + \nu\nu^{\top})\right\| \leqslant \frac{p\sqrt{\log(1/\rho)}}{\sqrt{n}}$$

In particular, for $n$ large enough, $\lambda_{\min}(\frac{1}{n_{\mathrm{tr}}}Z_{\mathrm{tr}}^{\top}Z_{\mathrm{tr}}) \geqslant 1/2$.

Similarly,

$$\frac{1}{n_{\mathrm{tr}}}Z_{\mathrm{tr}}^{\top}Y_{\mathrm{tr}} = M^{\top}\left(\frac{n_1}{n_{\mathrm{tr}}}\frac{1}{n_1}\sum_{I_1} x_i - \frac{n_{-1}}{n_{\mathrm{tr}}}\frac{1}{n_{-1}}\sum_{I_{-1}} x_i\right)$$

Using the fact that $\|x\| = \sup_{\|u\|\leqslant 1} u^{\top}x$ and for such a $u$ the variable $u^{\top}(z - \nu)$ is unit Gaussian, applying Lemma 6 we get that with probability $1 - \rho$

$$\left\|\frac{1}{n_1}\sum_{I_1} z_i - \nu\right\| \lesssim \frac{\sqrt{p} + \sqrt{\log(1/\rho)}}{\sqrt{n}}$$

and similarly for $I_{-1}$, and therefore

$$\left\|\frac{1}{n_{\mathrm{tr}}}Z_{\mathrm{tr}}^{\top}Y_{\mathrm{tr}} - \nu\right\| \leqslant \frac{\sqrt{p} + \sqrt{\log(1/\rho)}}{\sqrt{n}}$$

At the end of the day

$$\left\| \hat{\beta} - ((\lambda+1)\mathrm{Id} + \nu\nu^\top)^{-1}\nu \right\| \leqslant \left\| (\lambda\mathrm{Id} + Z_{\mathrm{tr}}^\top Z_{\mathrm{tr}}/n)^{-1}(Z_{\mathrm{tr}}^\top Y_{\mathrm{tr}}/n - \nu) \right\|$$

$$+ \left\| ((\lambda\mathrm{Id} + Z_{\mathrm{tr}}^\top Z_{\mathrm{tr}}/n_{\mathrm{tr}})^{-1} - ((\lambda+1)\mathrm{Id} + \nu\nu^\top)^{-1})\nu \right\|$$

$$\lesssim \frac{\sqrt{p} + \sqrt{\log(1/\rho)}}{\sqrt{n}} + \frac{\|\nu\|}{\lambda} \left\| Z_{\mathrm{tr}}^\top Z_{\mathrm{tr}}/n - (\mathrm{Id} + \nu\nu^\top) \right\|$$

$$\lesssim \frac{\|\nu\|\, p\sqrt{\log(1/\rho)}}{\sqrt{n}}$$

Moreover, $\nu$ is an eigenvector for $(\lambda+1)\mathrm{Id} + \nu\nu^\top$ so the limit is actually:

$$\hat{\beta}_{lim} = ((\lambda+1)\mathrm{Id} + \nu\nu^\top)^{-1}\nu = \frac{\nu}{1 + \lambda + \|\nu\|^2}$$

Let us now compute the limit of the test risk:

$$\frac{1}{n_{\mathrm{te}}} \left\| Y_{\mathrm{te}} - Z_{\mathrm{te}}^\top \hat{\beta} \right\|^2 = 1 - 2\hat{\beta}^\top \left( \frac{1}{n_{\mathrm{te}}} Z_{\mathrm{te}}^\top Y_{\mathrm{te}} \right) + \hat{\beta}^\top \left( \frac{1}{n_{\mathrm{te}}} Z_{\mathrm{te}}^\top Z_{\mathrm{te}} \right) \hat{\beta}^\top$$

By a reasoning identical to the one above on $\frac{1}{n_{\mathrm{te}}} Z_{\mathrm{te}}^\top Y_{\mathrm{te}} \approx \nu$ and $\frac{1}{n_{\mathrm{te}}} Z_{\mathrm{te}}^\top Z_{\mathrm{te}} \approx \mathrm{Id} + \nu\nu^\top$, and using $\left\| \hat{\beta}_{lim} \right\| \leqslant \|\nu\|$ and $\left\| \mathrm{Id} + \nu\nu^\top \right\| = 1 + \|\nu\|^2$, we obtain

$$\frac{1}{n_{\mathrm{te}}} \left\| Y_{\mathrm{te}} - Z_{\mathrm{te}}^\top \hat{\beta} \right\|^2 = 1 - 2(\hat{\beta}_{lim})^\top \nu + (\hat{\beta}_{lim})^\top \left( \mathrm{Id} + \nu\nu^\top \right) \hat{\beta}_{lim} + \mathcal{O}\left( \frac{\|\nu\|^4\, p\sqrt{\log(1/\rho)}}{\sqrt{n}} \right)$$

$$= 1 - 2\frac{\|\nu\|^2}{1 + \lambda + \|\nu\|^2} + \frac{\|\nu\|^2 + \|\nu\|^4}{(1 + \lambda + \|\nu\|^2)^2} + \mathcal{O}\left( \frac{\|\nu\|^4\, p\sqrt{\log(1/\rho)}}{\sqrt{n}} \right)$$

$$= \frac{(1+\lambda)^2 + \|\nu\|^2}{(1 + \lambda + \|\nu\|^2)^2} + \mathcal{O}\left( \frac{\|\nu\|^4\, p\sqrt{\log(1/\rho)}}{\sqrt{n}} \right)$$

We use a union bound over all these inequalities to conclude the proof. $\qquad\square$

## B.2  Proof of Theorem 7

We start with the proof of Lemma 2.

*Proof of Lemma 2.* The proof is similar to that of Lemma 1. Here we denote by $I_1, I_{-1} \subset \{1, \ldots, n\}$ the indices of the $x_i$ from the first and second community, of size $n_1$ and $n_{-1}$, from the whole sample set. Again, since the communities are balanced, with probability $1 - \rho$ we have $|I_1|/n \approx \frac{1}{2} + \mathcal{O}\left( \sqrt{\log(1/\rho)/n} \right)$.

We decompose

$$\|\tilde{x}_i - \varphi_{\mathrm{cl.}}(x_i)\| = \left\| \frac{\varepsilon \sum_j x_j + \sum_j W_g(x_i,x_j)x_j}{n\varepsilon + \sum_j W_g(x_i,x_j)} - \varphi_{\mathrm{cl.}}(x_i) \right\|$$

$$= \left\| \frac{\varepsilon \frac{1}{n}\sum_j x_j + \frac{1}{n}\sum_j W_g(x_i,x_j)x_j}{\varepsilon + \frac{1}{n}\sum_j W_g(x_i,x_j)} - \frac{\frac{1}{2}\left(d_\mu(x_i)\left(\frac{x_i+\mu}{2}\right) + d_{-\mu}(x_i)\left(\frac{x_i-\mu}{2}\right)\right)}{\varepsilon + \frac{1}{2}(d_\mu(x_i) + d_{-\mu}(x_i))} \right\|$$

$$\leqslant \frac{1}{n}\left\| \sum_j x_j \right\| + \left\| \frac{\frac{1}{n}\sum_j W_g(x_i,x_j)x_j}{\varepsilon + \frac{1}{n}\sum_j W_g(x_i,x_j)} - \frac{\frac{1}{2}\left(d_\mu(x_i)\left(\frac{x_i+\mu}{2}\right) + d_{-\mu}(x_i)\left(\frac{x_i-\mu}{2}\right)\right)}{\varepsilon + \frac{1}{n}\sum_j W_g(x_i,x_j)} \right\|$$

$$+ \left\| \frac{\frac{1}{2}\left(d_\mu(x_i)\left(\frac{x_i+\mu}{2}\right) + d_{-\mu}(x_i)\left(\frac{x_i-\mu}{2}\right)\right)}{\varepsilon + \frac{1}{n}\sum_j W_g(x_i,x_j)} - \frac{\frac{1}{2}\left(d_\mu(x_i)\left(\frac{x_i+\mu}{2}\right) + d_{-\mu}(x_i)\left(\frac{x_i-\mu}{2}\right)\right)}{\varepsilon + \frac{1}{2}(d_\mu(x_i) + d_{-\mu}(x_i))} \right\|$$

$$\leqslant \frac{1}{n}\left\| \sum_j x_j \right\| + \varepsilon^{-1}\left\| \frac{1}{n}\sum_j W_g(x_i,x_j)x_j - \frac{1}{2}\left(d_\mu(x_i)\left(\frac{x_i+\mu}{2}\right) + d_{-\mu}(x_i)\left(\frac{x_i-\mu}{2}\right)\right) \right\|$$

$$+ \frac{1}{4\varepsilon^2}\left| \frac{1}{n}\sum_j W_g(x_i,x_j) - \frac{d_\mu(x_i) + d_{-\mu}(x_i)}{2} \right| \left\| d_\mu(x_i)\left(\frac{x_i+\mu}{2}\right) + d_{-\mu}(x_i)\left(\frac{x_i-\mu}{2}\right) \right\|$$

For the first term, with probability $1 - \rho$ we have

$$\frac{1}{n}\left\|\sum_j x_j\right\| = \frac{n_1}{n}\frac{1}{n_1}\left\|\sum_{j\in I_1} x_j - \mu\right\| + \frac{n_{-1}}{n}\frac{1}{n_{-1}}\left\|\sum_{j\in I_{-1}} x_j + \mu\right\|$$

$$\leqslant \left\|\frac{1}{n_1}\sum_{j\in I_1} x_j - \mu\right\| + \left\|\frac{1}{n_{-1}}\sum_{j\in I_{-1}} x_j + \mu\right\|$$

$$\lesssim \frac{\log n\left(\sqrt{d} + \sqrt{\log(1/\rho)}\right)}{\sqrt{n}}$$

Where we have used Lemma 7 with $W = 1$ and $n_1 = n - n_{-1} \approx n/2$ for the last line.

For the second term, similarly

$$\left\|\frac{1}{n}\sum_j W_g(x_i, x_j)x_j - \frac{1}{2}\left(d_\mu(x_i)\left(\tfrac{x_i+\mu}{2}\right) + d_{-\mu}(x_i)\left(\tfrac{x_i-\mu}{2}\right)\right)\right\|$$

$$\lesssim \frac{1}{2}\left\|\frac{1}{n_1}\sum_{j\in I_1} W_g(x_i, x_j)x_j - d_\mu(x_i)\left(\frac{x_i+\mu}{2}\right)\right\|$$

$$+ \frac{1}{2}\left\|\frac{1}{n_{-1}}\sum_{j\in I_{-1}} W_g(x_i, x_j)x_j - d_{-\mu}(x_i)\left(\frac{x_i-\mu}{2}\right)\right\| + \sqrt{\log(1/\rho)/n}$$

Using Lemma 5, we have $\mathbb{E}_\mathcal{N} W_g(x, X)X = d_0(x)\frac{x}{2}$. Hence, using the Lipschitz properties of the Gaussian kernel, and since we can center

$$\frac{1}{n_1}\sum_{j\in I_1} W_g(x_i, x_j)x_j - d_\mu(x_i)\left(\frac{x_i+\mu}{2}\right) = \frac{1}{n_1}\sum_{j\in I_1} W_g(x_i-\mu, x_j-\mu)(x_j-\mu) - d_\mu(x_i-\mu)\left(\frac{x_i-\mu}{2}\right)$$

and $x_j - \mu \sim \mathcal{N}$, using Lemma 7 we obtain

$$\left\|\frac{1}{n_1}\sum_{j\in I_1} W_g(x_i, x_j)x_j - d_\mu(x_i)\left(\frac{x_i+\mu}{2}\right)\right\| \lesssim \frac{\log n\left(\sqrt{d} + \sqrt{\log(1/\rho)}\right)}{\sqrt{n}}$$

We proceed similarly for the second term, and again with the first part of Lemma 7 to obtain

$$\left|\frac{1}{n}\sum_j W_g(x_i, x_j) - \frac{d_\mu(x_i) + d_{-\mu}(x_i)}{2}\right| \lesssim \frac{\sqrt{\log n}\left(\sqrt{d} + \sqrt{\log(1/\rho)}\right)}{\sqrt{n}}$$

which gives us the first result. The second is obtained by simply decomposing and using $\|\varphi_{\text{cl.}}(x)\| \lesssim 1/\varepsilon$. $\qquad\square$

We will need the following Lemma, similar to Lemma 3.

**Lemma 4.** *Let* $x \sim \mathcal{N}_\mu$. *We have*

$$\|\mathbb{E}\varphi_{\text{cl.}}(x) - \mu\| \lesssim \sqrt{d}2^{d/2}\|\mu\|\varepsilon^{1/4} + \frac{\|\mu\|}{\varepsilon^3}e^{-\|\mu\|^2/4}$$

$$\left\|\mathbb{E}\varphi_{\text{cl.}}(x)\varphi_{\text{cl.}}(x)^\top - (\mu\mu^\top + \text{Id}/4)\right\| \lesssim d2^{d/2}\|\mu\|^2\varepsilon^{1/4} + \frac{\|\mu\|^2\sqrt{d}}{\varepsilon^3}e^{-\|\mu\|^2/4}$$

*Proof.* Denote by $\mathcal{B}_{\mu,r}$ a ball of radius $r$ around $\mu$. Within this ball, $d_\mu(x) \geqslant 2^{-d/2}e^{-r^2/4}$, while $d_{-\mu}(x) \leqslant 2^{-d/2}e^{-\|\mu\|^2/4}$. We also recall that $\int_{\mathcal{B}^c_{\mu,r}} \mathcal{N}_\mu \leqslant e^{-r^2/2}$ and $\int_{\mathcal{B}^c_{\mu,r}} \|x - \mu\|^2\mathcal{N}_\mu \lesssim$

$2^{d/2}e^{-r^2/4}$. Now we decompose

$$E_{\mathcal{N}_\mu}\left\|\varphi_{\text{cl.}}(x) - \frac{x+\mu}{2}\right\|^2 = \int \left\|\frac{d_\mu(x)\left(\frac{x+\mu}{2}\right) + d_{-\mu}(x)\left(\frac{x-\mu}{2}\right)}{2\varepsilon + d_\mu(x) + d_{-\mu}(x)} - \frac{x+\mu}{2}\right\|^2 \mathcal{N}_\mu(x)dx$$

$$= \int \left\|\frac{\varepsilon(x+\mu) - d_{-\mu}(x)\mu}{2\varepsilon + d_\mu(x) + d_{-\mu}(x)}\right\|^2 \mathcal{N}_\mu(x)dx$$

$$\lesssim \int_{\mathcal{B}_{\mu,r}} \left\|\frac{\varepsilon(x+\mu) - d_{-\mu}(x)\mu}{2\varepsilon + d_\mu(x) + d_{-\mu}(x)}\right\|^2 \mathcal{N}_\mu(x)dx$$

$$+ \int_{\mathcal{B}_{-\mu,r}} \left\|\frac{\varepsilon(x+\mu) - d_{-\mu}(x)\mu}{2\varepsilon + d_\mu(x) + d_{-\mu}(x)}\right\|^2 \mathcal{N}_\mu(x)dx$$

$$+ \int_{(\mathcal{B}_{\mu,r} \cup \mathcal{B}_{-\mu,r})^c} \left\|\frac{\varepsilon(x+\mu) - d_{-\mu}(x)\mu}{2\varepsilon + d_\mu(x) + d_{-\mu}(x)}\right\|^2 \mathcal{N}_\mu(x)dx$$

$$\lesssim \varepsilon^2(\|\mu\|^2 + d)2^d e^{-r^2/2} + \|\mu\|^2 e^{-\|\mu\|^2/2}e^{r^2/2}$$

$$+ (\varepsilon^2 2^d \|\mu\| e^{r^2/4} + \|\mu\|^2)e^{-\|\mu\|^2/4}$$

$$+ 2^d e^{-r^2/4}\|\mu\|^2 + \|\mu\|^2 2^d e^{-r^2/2}e^{-r^2/2}/\varepsilon^2$$

$$\lesssim d2^d \|\mu\|^2 \sqrt{\varepsilon} + \frac{\|\mu\|^2}{\varepsilon^{3/2}}e^{-\|\mu\|^2/2}$$

Where the last line is obtained by choosing $r = \sqrt{3\log(1/\varepsilon)}$. Then we use that for two random variables $X$ and $Y$, $\|\mathbb{E}X - \mathbb{E}Y\| \leqslant \sqrt{\mathbb{E}\|X-Y\|^2}$, and $\|\mathbb{E}XX^\top - \mathbb{E}YY^\top\| \leqslant (\sqrt{\mathbb{E}\|X\|^2} + \sqrt{\mathbb{E}\|Y\|^2})\sqrt{\mathbb{E}\|X-Y\|^2}$ to conclude. $\square$

We are now ready to prove Theorem 7.

*Proof of Theorem 7.* We proceed as in the proof of Theorem 4. We define $\hat{\beta}^\varphi$ and $\mathcal{R}^\varphi$ by replacing $Z^{(k)}$ with $Z^\varphi = X^\varphi M$ where the rows of $X^\varphi$ are the $\varphi_{\text{cl.}}(x_i)$.

Since $\|\varphi_{\text{cl.}}(x)\| \leqslant \max\left(\frac{\|x+\mu\|}{2}, \frac{\|x-\mu\|}{2}\right)$, $\varphi_{\text{cl.}}(x)$ is a subgaussian vector with $\|u^\top \varphi_{\text{cl.}}(x)\|_{\psi_2} \lesssim \|u^\top x\|_{\psi_2} \lesssim 1$. We can therefore apply the same reasoning as in the proof of Theorem 6 and concentrate $(Z_{\text{tr}}^\varphi)^\top Z_{\text{tr}}^\varphi / n_{\text{tr}}$. Using Lemma 4, for $\varepsilon$ small enough, and $\|\mu\|$ and $n$ large enough, it is almost $\text{Id}/4 + \nu\nu^\top$, and thus $\lambda_{\min}((Z_{\text{tr}}^\varphi)^\top Z_{\text{tr}}^\varphi / n_{\text{tr}}) \gtrsim 1$. Finally, using Lemma 2, for $n_{\text{tr}}$ large enough $\lambda_{\min}((Z_{\text{tr}}^{(1)})^\top Z_{\text{tr}}^{(1)} / n_{\text{tr}}) \geqslant \lambda_{\min}((Z_{\text{tr}}^\varphi)^\top Z_{\text{tr}}^\varphi / n_{\text{tr}})/2 \gtrsim 1$.

Since $\|\varphi_{\text{cl.}}(x)\| \leqslant 1/\varepsilon$, using Lemma 2 we bound

$$\left\|\hat{\beta} - \hat{\beta}^\varphi\right\| = \left\|(\lambda\text{Id} + (Z_{\text{tr}}^{(1)})^\top Z_{\text{tr}}^{(1)}/n_{\text{tr}})^{-1}(Z_{\text{tr}}^{(1)})^\top Y_{\text{tr}}/n_{\text{tr}} - (\lambda\text{Id} + (Z_{\text{tr}}^\varphi)^\top Z_{\text{tr}}^\varphi/n_{\text{tr}})^{-1}(Z_{\text{tr}}^\varphi)^\top Y_{\text{tr}}/n_{\text{tr}}\right\|$$

$$\leqslant \left\|((\lambda\text{Id} + (Z_{\text{tr}}^{(1)})^\top Z_{\text{tr}}^{(1)}/n_{\text{tr}})^{-1} - (\lambda\text{Id} + (Z_{\text{tr}}^\varphi)^\top Z_{\text{tr}}^\varphi/n_{\text{tr}})^{-1})(Z_{\text{tr}}^\varphi)^\top Y_{\text{tr}}/n_{\text{tr}}\right\|$$

$$+ \left\|(\lambda\text{Id} + (Z_{\text{tr}}^{(1)})^\top Z_{\text{tr}}^{(1)}/n_{\text{tr}})^{-1}((Z_{\text{tr}}^{(1)})^\top Y_{\text{tr}}/n_{\text{tr}} - (Z_{\text{tr}}^\varphi)^\top Y_{\text{tr}}/n_{\text{tr}})\right\|$$

$$\leqslant \frac{1}{\varepsilon}\sup_i \left\|x_i^{(1)}(x_i^{(1)})^\top - \varphi(x_i)\varphi(x_i)^\top\right\| + \sup_i \left\|x_i^{(1)} - \varphi(x_i)\right\|$$

$$\lesssim \frac{\text{poly}(1/\varepsilon)\log n(\sqrt{d} + \sqrt{\log(1/\rho)})}{\sqrt{n}}$$

Using the same bounds on $\frac{1}{n_{\text{te}}}(Z_{\text{te}}^{(1)})^\top Y_{\text{te}}$ and $\frac{1}{n_{\text{te}}}(Z_{\text{te}}^{(1)})^\top Z_{\text{te}}^{(1)}$, and using $\left\|\hat{\beta}^\varphi\right\| \lesssim \varepsilon^{-1}$, we get

$$\left|\mathcal{R}^{(1)} - \mathcal{R}^\varphi\right| \lesssim \frac{\text{poly}(1/\varepsilon)\log n(\sqrt{d} + \sqrt{\log(1/\rho)})}{\sqrt{n}} \tag{23}$$

Then, we apply the same reasoning as in the proof of Theorem 6, we obtain

$$\mathcal{R}^\varphi = 1 - 2(\mathbb{E}yz^\varphi)^\top \beta_{lim}^\varphi + (\beta_{lim}^\varphi)^\top \mathbb{E}z^\varphi(z^\varphi)^\top \beta_{lim}^\varphi + \mathcal{O}\left(\text{poly}(\frac{1}{\varepsilon}, \frac{1}{\lambda}) \frac{\log n(\sqrt{d} + \sqrt{\log(1/\rho)})}{\sqrt{n}}\right)$$

where $\beta_{lim}^\varphi = (\lambda \text{Id} + \mathbb{E}z^\varphi(z^\varphi)^\top)^{-1}(\mathbb{E}yz^\varphi)$.

Finally, using Lemma 4, and computations similar to 6, we obtain

$$\mathcal{R}^\varphi = R_{\text{cl.}}(1/4) + \mathcal{O}\left(\text{poly}(\|\mu\|, e^d)\left(\varepsilon^{1/4} + \frac{1}{\varepsilon^3}e^{-\|\mu\|^2/4}\right)\right)$$

which concludes the proof. □

## C   Technical Lemmas

This section gather some technical Lemmas used throughout the proofs. We start by some derivations on Gaussian distributions, then details the chaining concentration inequalities used in this work.

### C.1   Properties of Gaussians

**Lemma 5** (Gaussian integral). *Let* $W(x, y) = e^{-\frac{1}{2}\|x-y\|_{\Sigma_W^{-1}}}$ *be the Gaussian kernel with covariance* $\Sigma_W$. *We have*

$$d(x) := \int W(x, y)\mathcal{N}_{\mu,\Sigma}(y)dy = \frac{|\Sigma_W|^{\frac{1}{2}}}{|\Sigma_W + \Sigma|^{\frac{1}{2}}}e^{-\frac{1}{2}\|x-\mu\|^2_{(\Sigma_W+\Sigma)^{-1}}} \tag{24}$$

$$\mathcal{L}(x) := \int W(x, y)y\mathcal{N}_{\mu,\Sigma}(y)dy = d(x)(\Sigma_W^{-1} + \Sigma^{-1})^{-1}(\Sigma_W^{-1}x + \Sigma^{-1}\mu) \tag{25}$$

*Proof.* We have the following when $n \to \infty$.

$$d(x) = \int W(x, y)\mathcal{N}_{\mu,\Sigma}(y)dy = \int e^{-\frac{1}{2}\|x-y\|^2_{\Sigma_W^{-1}}}\mathcal{N}_{\mu,\Sigma}(y)dy$$

$$= (2\pi)^{d/2}|\Sigma_W|^{\frac{1}{2}}\int \mathcal{N}_{0,\Sigma_W}(x - y)\mathcal{N}_{\mu,\Sigma}(y)dy$$

$$= (2\pi)^{d/2}|\Sigma_W|^{\frac{1}{2}}\mathcal{N}_{0,\Sigma_W} \star \mathcal{N}_{\mu,\Sigma}(x)$$

$$= (2\pi)^{d/2}|\Sigma_W|^{\frac{1}{2}}\mathcal{N}_{\mu,\Sigma_W+\Sigma}(x) = \frac{|\Sigma_W|^{\frac{1}{2}}}{|\Sigma_W + \Sigma|^{\frac{1}{2}}}e^{-\frac{1}{2}\|x-\mu\|^2_{(\Sigma_W+\Sigma)^{-1}}}$$

Since the convolution of two gaussians is a Gaussian. And

$$\mathcal{L}(x) = \int W(x, y)y\mathcal{N}_{\mu,\Sigma}(y)dy$$

$$= \frac{1}{(2\pi)^{d/2}|\Sigma|^{\frac{1}{2}}}\int ye^{-\frac{1}{2}\|y-x\|^2_{\Sigma_W^{-1}} - \frac{1}{2}\|y-\mu\|^2_{\Sigma^{-1}}}dy$$

$$= \frac{1}{(2\pi)^{d/2}|\Sigma|^{\frac{1}{2}}}\int -(\Sigma_W^{-1}(y-x) + \Sigma^{-1}(y-\mu))e^{-\frac{1}{2}\|y-x\|^2_{\Sigma_W^{-1}} - \frac{1}{2}\|y-\mu\|^2_{\Sigma^{-1}}}dy$$

$$+ (\text{Id} + \Sigma_W^{-1} + \Sigma^{-1})\mathcal{L}(x) - (\Sigma_W^{-1}x + \Sigma^{-1}\mu)d(x)$$

$$= d(x)(\Sigma_W^{-1} + \Sigma^{-1})^{-1}(\Sigma_W^{-1}x + \Sigma^{-1}\mu)$$

using that the first term in the sum is 0 since it is the integral of a derivative. □

## C.2 Chaining and subgaussian variables

A random variable $X$ is said to be *subgaussian* if

$$\|X\|_{\psi_2} := \inf\{t > 0; \mathbb{E}e^{X^2/t^2} \leqslant 2\} < \infty \tag{26}$$

A good reference on subgaussian random variables is [42, Chap. 2]. For a bounded random variable $X$ and subgaussian $Y$, we have immediately from the definition

$$\|XY\|_{\psi_2} \leqslant \|X\|_{\infty} \|Y\|_{\psi_2} \tag{27}$$

**Lemma 6** (Chaining on non-normalized kernels). *Consider $x_i \sim P \in \mathcal{P}(\mathbb{R}^m)$, a ball $\mathcal{B}_r \subset \mathbb{R}^n$ with respect to a metric $d$, and a bivariate function $F : \mathbb{R}^n \times \mathbb{R}^m \to \mathbb{R}$ that satisfies:*

1. *For all $z \in \mathcal{B}_r$, $F(z, X)$ is subgaussian with norm $\|F(z, X)\|_{\psi_2} \leqslant C$*

2. *For all $z, z' \in \mathcal{B}_r$, $\|F(z, X) - F(z', X)\|_{\psi_2} \leqslant C_L d(z, z')$*

*Then, with probability at least $1 - \rho$,*

$$\sup_{z \in \mathcal{B}_r} \left| \frac{1}{n} \sum_i F(z, x_i) - \int F(z, x) dP(x) \right|_{\infty} \lesssim \frac{rC_L \left(\sqrt{d} + \sqrt{\log(1/\rho)}\right) + C\sqrt{\log(1/\rho)}}{\sqrt{n}}$$

*Proof.* Define

$$Y_z = \frac{1}{n} \sum_i F(z, x_i) - \int F(z, x) dP(x)$$

By [42, Lemma 2.6.8], we have $\|Y_z\|_{\psi_2} \leqslant C$. Hence we can apply a generalized Hoeffding's inequality for subgaussian variables: with probability at least $1 - \rho$,

$$|Y_{z_0}| \lesssim \frac{C\sqrt{\log(1/\rho)}}{\sqrt{n}}$$

For any $z_0$, we have

$$\sup_{z \in \mathcal{B}_r} |Y_z| \leqslant \sup_{z, z' \in \mathcal{B}_r} |Y_z - Y_{z'}| + |Y_{z_0}|$$

The second term is bounded by the inequality above. For the first term, we are going to use Dudley's inequality "tail bound" version [42, Thm 8.1.6]. We first need to check the sub-gaussian increments of the process $Y_z$. For any $z, z' \in \mathcal{B}_r$, we have

$$\|Y_z - Y_{z'}\|_{\psi_2} \lesssim \frac{1}{n} \left( \sum_{i=1}^{n} \|(F(z, x_i) - F(z', x_i)) - \mathbb{E}((F(z, X) - F(z', X)))\|_{\psi_2}^2 \right)^{\frac{1}{2}}$$

$$\lesssim \frac{1}{n} \left( \sum_{i=1}^{n} \|(F(z, x_i) - F(z', x_i))\|_{\psi_2}^2 \right)^{\frac{1}{2}}$$

$$\leqslant \frac{C_L}{\sqrt{n}} d(z, z')$$

where we have used, from [42], Prop. 2.6.1 for the first line, Lemma 2.6.8 for the second, and the properties of $F$ for the last.

Now, we apply Dudley's inequality [42, Thm 8.1.6] to obtain that with probability $1 - \rho$,

$$\sup_{z, z' \in \mathcal{B}_r} |Y_z - Y_{z'}| \lesssim \frac{C_L}{\sqrt{n}} \left( \int_0^r \sqrt{\log N(\mathcal{B}_r, d, \varepsilon)} d\varepsilon + \sqrt{\log(1/\rho)} r \right)$$

$$\lesssim \frac{C_L r}{\sqrt{n}} \left( \sqrt{d} + \sqrt{\log(1/\rho)} \right)$$

which concludes the proof. $\qquad\square$

**Lemma 7.** *Let $x_1, \ldots, x_n$ be iid $\mathcal{N}_{0,\Sigma}$ on $\mathbb{R}^d$, and $W$ be a $1$-bounded, $C$-Lipschitz kernel in the first variable with respect to the metric $\|\cdot\|_{\Sigma^{-1}}$.*

*With probability at least $1 - \rho$,*

$$\sup_i \left| \frac{1}{n} \sum_j W(x_i, x_j) - \mathbb{E}W(x_i, X) \right| \lesssim \frac{\sqrt{\log n} \, C\left(\sqrt{d} + \sqrt{\log(1/\rho)}\right)}{\sqrt{n}} \tag{28}$$

*With probability at least $1 - \rho$,*

$$\sup_i \left\| \frac{1}{n} \sum_j W(x_i, x_j)x_j - \mathbb{E}W(x_i, X)X \right\|_{\Sigma^{-1}} \lesssim \frac{\log n \, C\left(\sqrt{d} + \sqrt{\log(1/\rho)}\right)}{\sqrt{n}} \tag{29}$$

*Proof.* By the properties of Gaussian variables and a union bound, with probability at least $1 - \rho$,

$$\forall i, \|x\|_{\Sigma^{-1}} \lesssim \sqrt{\log n} =: r_n \tag{30}$$

Now, since $W$ is bounded, $W(x, X)$ is subgaussian for any $x$. Applying Lemma 6 with $F = W$ and considering that $\|\cdot\|_{\psi_2} \leqslant \|\cdot\|_\infty$, we get that with probability at least $1 - \rho$,

$$\sup_{\|x\|_{\Sigma^{-1}} \leqslant r_n} \left| \frac{1}{n} \sum_j W(x, x_j) - \mathbb{E}W(x, X) \right| \lesssim \frac{r_n C\left(\sqrt{d} + \sqrt{\log(1/\rho)}\right)}{\sqrt{n}} \tag{31}$$

Combining with (30), we get (28).

Now, we write

$$\sup_{\|x\|_{\Sigma^{-1}} \leqslant r_n} \left\| \frac{1}{n} \sum_j W(x, x_j)x_j - \int W(x, x')x' \mathcal{N}_{0,\Sigma}(x')dx' \right\|_{\Sigma^{-1}}$$

$$= \sup_{\|x\|_{\Sigma^{-1}} \leqslant r_n} \sup_{\|u\|_\Sigma \leqslant 1} \left| \frac{1}{n} \sum_j W(x, x_j)u^\top x_j - \int W(x, x')u^\top x' \mathcal{N}_{0,\Sigma}(x')dx' \right|$$

We aim to apply again Lemma 6 for the function $F((x, u), x') = W(x, x')u^\top x'$ and the metric $\|x\|_{\Sigma^{-1}} + \|u\|_\Sigma$. First, for any $u$ with $\|u\|_\Sigma \leqslant 1$, $u^\top X$ is Gaussian with variance less than 1, so $\|W(x, X)u^\top X\|_{\psi_2} \leqslant \|W(x, \cdot)\|_\infty \|u^\top X\|_{\psi_2} \lesssim 1$. Similarly, $(u - u')^\top X$ is Gaussian with variance $\|u - u'\|_\Sigma$, so

$$\|F((x, u), X) - F((x', u'), X)\|_{\psi_2} \leqslant \|W(x, \cdot) - W(x', \cdot)\|_\infty \|(u - u')^\top X\|_{\psi_2}$$
$$\lesssim C \|x - x'\|_{\Sigma^{-1}} \|u - u'\|_\Sigma$$
$$\lesssim r_n C(\|x - x'\|_{\Sigma^{-1}} + \|u - u'\|_\Sigma)$$

Hence, we get

$$\sup_{\|x\|_{\Sigma^{-1}} \leqslant r_n} \left\| \frac{1}{n} \sum_j W(x, x_j)x_j - \mathbb{E}W(x, X)X \right\| \lesssim \frac{r_n^2 C\left(\sqrt{d} + \sqrt{\log(1/\rho)}\right)}{\sqrt{n}} \tag{32}$$

which concludes the proof. $\square$