# OpenReview forum: "Not too little, not too much: a theoretical analysis of graph (over)smoothing"
_NeurIPS.cc/2022/Conference — NeurIPS 2022 Accept_

### Official Review · Reviewer_hUnE · 2022-06-22

**Rating:** 4
**Confidence:** 4
**Soundness:** 4 excellent
**Presentation:** 4 excellent
**Contribution:** 2 fair

**Summary:**

This paper proposes a simplified theoretical framework -- linear MPNNs with mean aggregation, latent variables underlying the label, feature, and graph generation -- to show that while the problem of over-smoothing is indeed generally relevant in the context of graph neural networks, a finite number of aggregation steps (equivalently finite integration time from the point of view of GNNs as dynamical systems) could be desirable to reduce the noise of the node representations and consequently improve the classification.

**Questions:**

Questions and further feedback:

- Is Assumption 1 in some way optimal or a consequence of the techniques adopted (e.g. that we show that in probability $\mathcal{R}^{(1)} < \mathcal{R}^{(0)}$?

- It would be more interesting to explore what happens if we have a residual connection which is known to mitigate (prevent) over-smoothing. How would now the behaviour of the risk be affected? Any speculation?

- What of the analysis proposed here can be extended to less ideal scenarios? For example when the graph generation is less ideal than that encoded by a Gaussian kernel over the latent variables -- certainly the estimates are less quantitative but perhaps more interesting.

- Some more synthetic experiments could also potentially improve the story especially to complement qualitative analysis when stress-testing the ideal scenario. For example, what happens when we generate features (or graph) that fail to satisfy the assumptions? How would the degradation wrt k react?

Conclusions:

It is hard to have a strong opinion on this paper. On one side, it is very well written and presented, with the proposed problem fully addressed in the specific mathematical setting introduced. While ideal scenarios are indeed useful for providing a better understanding of a given phenomenon, I can't help but feel that the insights deduced here are not that novel. In fact, I circle back to the sentence "(...) and we hope that it proves an inspiration to help the design of GNNs in the future", $\textbf{which has not been justified in my opinion}$. Nonetheless, I still think $\textit{this is a worthy paper}$ -- with potential interesting extensions in some of the less ideal cases I mentioned earlier -- $\textit{perhaps though more suitable for a more `statistics oriented/flavoured' conference rather than a general one}$. I am of course happy to engage in discussion and raise my score if the authors and/or other reviewers point to something I have missed in my analysis concerning impact and relevance in the community.

**Limitations:**

Societal impact is not reported -- albeit the paper is theoretical in nature. Limitations are properly discussed.

**Strengths And Weaknesses:**

Presentation: The paper is well written and the outline is clear. The problem at hand is precisely formalized along with main results and ingredients entering the proofs. I also appreciate the discussion on limitations of the current framework and/or techniques of the adopted proofs.

Novelty and originality: The paper is original -- at least in the GNN community -- for what concerns the proposed risk analysis as a function of the number of aggregation steps. In terms of "novelty", the judgement is slightly trickier. Per se, the paper does not really offer new "carry-with-you" insights and there is perhaps little new understanding beyond an elegant mathematical analysis that only applies to very ideal scenarios though -- as further commented below. The following specific sentence at line 74 is crucial: "(...) the relationship between node features and graph structure, and we hope that it proves an inspiration to help the design of GNNs in the future." It is not clear to me how and to what extent the paper provides an original insight to the community that could lead to better designed GNNs or more generally to a deeper understanding of their mechanism -- beyond better formalizing something that is arguably already known.

Strengths:

- The paper is elegant and no clear improvements are needed on the presentation front.

- The technical statements seem correct to me -- you may need an extra factor of 1/2 in front of the regularization in eq (3) for the equality to be correct?

- The problem addressed is worth exploring -- namely a finer analysis of GNNs after a $finite$ number of steps (i.e. finite integration time) is still somewhat lacking and the now standard asymptotic analysis of the over-smoothing effect is indeed limiting and ideal.

- I appreciate the efforts to comment on the technical assumption 1 and to provide some intuition based on principal components.

Weaknesses:

- The main weakness -- as partly mentioned above -- is that the message of the paper is known in the community and that the setting analysed does avoid investigating precisely those cases that are not fully understood. The upshot is the following: if the graph is homophilic (label distribution aligned with the topology) and the node-wise features are somewhat smooth (aware of the graph structure), then some (finite) smoothing is known and expected to be beneficial to denoise the node representations and leverage the information contained in the graph structure. This is for example implicit in all the references that argue how conventional graph convolutional models behave as low-pass filter operations in the graph-frequency domain (under some assumptions). While this may not have been formalized using the techniques of the paper, the end-message is not different. It is hard for me to grasp some novel understanding that the paper may bring in this scenario beyond some specific asymptotics that are though consequence of the very ideal setting analysed.

- Connected to the first point, the paper does analyse the easier problem of MPNNs "in the case of homophily". In fact, the underlying
assumption is that the graph generation procedure via the kernel is homophilic wrt the latent variables and hence $y = x^\top \beta^{\star}$. This is precisely the scenario where some steps of low-pass filters are beneficial and accordingly the same applies to MPNNs that act as average on the 1-hop. There is an implicit message throughout the paper that some smoothing is "always" beneficial. This may fail for heterophilic graphs when using MPNNs that simply average over the neighbours. This point should be better addressed.

- The relation between the optimal k and the dimension (which would be new in the community) is quite interesting but this is not explored further. While I understand that would require more involved mathematical techniques,  the behaviour of the risk after k steps and explicit formulas in low-dimension are only available after further constraints (for example on the form of $M$) and the far more interesting question of the degradation of k even in the ideal scenario is mostly still open given that the main theorem amounts to showing that with high probability one step of aggregation is better than none.

---

> ### Author Response · Authors · 2022-07-27
> **First rebuttal**
>
> We thank the reviewer for their very detailed and insightful feedback. We try to answer the different points as best as possible below.
>
> - **On novelty:**
> We fully agree that the main messages, both on beneficial smoothing and oversmoothing, are intuitive and already known by the community. Our main goal here was to rigorously bring together two messages that always feel separated in the existing (theoretical) literature: either GNNs need to be deep enough (to be powerful, and so on) with *no* limit to their depth (like in WL analysis), or deep architectures oversmooth. We still believe that some parts of the proposed analysis are somewhat original: the modelization of the graph with "missing" information in the node features, which really permits this analysis, and the few intuitions we were able to derive from our theoretical results. See below for more comments on the latter.
>
> - **On homophily/heterophily:**
> This is a very interesting point. It is true that the chosen graph model, although very classical in the literature, almost always results in homophilic graphs. However, we failed to mention (but will add in the revision) that this model does *not* always result in beneficial smoothing! In the 2d regression example for instance, if the regressor is in the direction of the small eigenvalue instead, then even one step of smoothing worsens the results, both theoretically and empirically. This case could be considered slightly "heterophilic", in the sense that close nodes (in the small eigenvalue direction) exhibit fast-varying labels. We will add this example in the paper.
> That said, beyond the somewhat artificial case described above, it is true that heterophily still quite mysterious in many ways. We believe that a realistic, non-trivial statistical modelization of heterophily is a necessary but still largely open question. Simply modelling node features as pure noise, or being uncorrelated, or inversely correlated, to the graph structure, is probably too simple. It is an outstanding open question, we will add a discussion in the paper.
>
> - **On the design of GNNs:**
> We agree that we should have elaborated more on this point, and apologize for not doing so. Since our analysis is largely based on "what shrinks and how fast", we were mostly thinking of the literature that combats oversmoothing with various *normalizations* processes, eg, "PairNorm: Tackling Oversmoothing in GNNs" (Zhao et al, ICLR2020), "Revisiting Over-smoothing in Deep GCNs" (Yang et al 2020), or "Supervised community detection with line graph neural networks" (Chen et al, ICLR2019, which includes a normalization step to mimick the power-method used in spectral clustering). In many of these works, the "normalization" can be somewhat arbitrary, and fights the collapsing of node features indiscriminately. However, if one were to better understand how graph smoothing shrinks various directions in the data or acts on communities, and how this might be beneficial or detrimental, then one may imagine more clever normalization strategies that would fight oversmoothing while maximizing the benefits of smoothing in particular directions. We agree that this is still largely open, but will add a discussion in the revision.
>
> - **On optimal $k$:**
> It is true that a rigorous proof of the optimal $k$, beyond the approximate expressions for $k>1$ that seem to match the numerics quite well, would be quite involved. The failure example described above may give a first example as to when smoothing is *not* beneficial, but other mechanisms are probably involved.
>
> **Questions:**
> - Asymptotically, $\mathcal{R}^{(1)}$ is the exact expression for the risk after *one* step of smoothing. So, the assumption is necessary and sufficient for the improvement brought by one step of smoothing. However, it is only *sufficient* for the existence of $k^\star>0$, since, as the error accumulate, there may be a $\mathcal{R}^{(k^\star)}<\mathcal{R}^{(0)}$ even if $\mathcal{R}^{(1)} > \mathcal{R}^{(0)}$ (we do not believe this to be very likely).
> - with residual connections, smoothed data are generally a convex combination of non-smoothed and fully smoothed data (in order to stay localized in space: say, $Z^{(k+1)} = ((1-\theta)Id + \theta L)Z^{(k)}$ ). In this case, the same conclusion would hold, as smoothing one step would bring improvement (but milder), and oversmoothing would kick in (but later). This would without doubt increase the optimal $k^\star$, but the effect on the optimal risk is not obvious. We will perform numerical tests and may include a section in the appendix.
> - if the edges are random bernoulli variable, we believe that all our results hold with the right concentration tools (eg "Sharp nonasymptotic bounds on the norm of random matrices with independent entries" by Bandeira et al), which we did not include for conciseness. We will perform more numerical experiments (especially on failure cases mentioned above) and include our results in the appendix.

---

> > ### Comment · Reviewer_hUnE · 2022-08-04
> > **Response to rebuttal**
> >
> > Thank you for your rebuttal.
> >
> > Unfortunately, I still believe that the novelty of the message is not strong enough. The setting is quite ideal and I am not convinced that general GNN experts would believe this work to lead to any new understanding or indeed new direction – or at least, I don’t see a significant new understanding or promising new directions.
> >
> > More specifically, beyond the simplified assumptions on labels and features, which almost always result in a homophilic graph which the community now understands to be a relatively easy task, the proposed framework reduces to collapsing a GNN into a single linear layer, which is an extremely simple setting that fails to apply to cases where we either have a residual connection or a non-linear activation. (Residual connection can prevent over-smoothing if used in a way other than a static global – i.e. homogeneous – convex combination).
> >
> > The paper also has essentially zero experiments. While I am personally of the idea that this is of course not a must, for a theoretical paper to get accepted in arguably the top ML venue some unknown results of general interest/applicability should be provided beyond formalizing existing intuitions under very nice assumptions. This is not done here and I still believe this paper, in its current state, should be targeting a more statistics’ flavoured venue where it should be a very easy “accept”. To use the words of another review, I actually can’t see what the “question of much current interest” is, since the much more interesting problem is not if smoothing helps when compared to no smoothing/infinite smoothing (using this model, with collapsed layers into a single linear one) but when/why the graph bias can be used and how (not necessarily smoothing) to lead to improved performances.
> >
> > Side note: I believe this paper is one of those cases where people might put it on the side of rejection or acceptance and there is no correct answer here. However, I don’t see how this paper - in its current state - satisfies the following definition “$\textbf{excellent impact}$ on at least one area, or high-to-excellent impact on multiple areas, with $\textbf{excellent evaluation}$”. I believe this is not the case and reviewers that instead argue for that have hardly put any concrete reason for why that holds.
> >  I have decided to stick to my score. Please note that this is a good paper and that my decision was not an easy one for me, but for all the reasons listed above, I think one needs a little more to target this general venue. It is also quite hard for me to gauge the other reviewers’ feedback given that we have two extremely high scores without any substantial arguments for that.

---

> > > ### Author Response · Authors · 2022-08-08
> > > **Small illustration of heterophily**
> > >
> > > Thank you for the additional comment. We agree with all limitations pointed out, which we find to be excellent inspiration for the future. Naturally we still believe the proposed theoretical study and proposed interpretations to be sufficient for a conference proceedings, and that further experiments, or novel practical designs of GNNs (and so on) would somewhat slightly dilute the message and all topics would not be properly treated in limited space.
> > >
> > > We however added a small illustration of heterophily and the failure of smoothing based on our linear regression study, as a new appendix in the supplementary material (App. D). It will be better integrated within the main body of the paper in a camera-ready version.

---

### Official Review · Reviewer_Xj6j · 2022-07-10

**Rating:** 8
**Confidence:** 2
**Soundness:** 4 excellent
**Presentation:** 4 excellent
**Contribution:** 4 excellent

**Summary:**

This paper studies the smoothing and over-smoothing phenomena on graph representation learning via the mean aggregation, which is the most common aggregation function of the message-passing framework. The authors prove that both phenomena are co-existent with a latent space random graph model. Besides, two mechanisms for the benefits of mean aggregation are recognized. In addition, the theoretical expressions are well matched with simulation results. The theory in this work has a great impact on graph representation learning. It helps researchers deeply comprehend the smoothing and over-smoothing phenomena. More importantly, this paper may enlighten researchers on preventing, relieving, and solving the over-smoothing issue.

**Questions:**

1. What are the boundary conditions between the smoothing and over-smoothing?

**Limitations:**

The authors have sufficiently addressed limitations of this work. In Section 6, the authors acknowledge that this work does not discuss the most common situation that how a non-linear GNN works for the node classification task with the cross entropy loss function.

**Strengths And Weaknesses:**

Strengths:

- The writing is neat and easy to follow.
- The theoretical foundation is solid.

---

> ### Author Response · Authors · 2022-07-27
> **First rebuttal**
>
> We thank the reviewer for their feedback.
>
> One question is raised by the reviewer, about the "boundary" between smoothing and over-smoothing. We believe this is related to the computation of the actual optimal $k^\star$, which may not be easy to do, especially in real-world scenarii. We hope nevertheless that this paper is a step toward deriving intuitions that may prove useful in more practical situations, eg, by building new estimations procedures for the various "directions" in which the graph structure provide useful smoothing of the data and incorporating this knowledge in various architectures.

---

### Official Review · Reviewer_seXB · 2022-07-11

**Rating:** 6
**Confidence:** 3
**Soundness:** 3 good
**Presentation:** 3 good
**Contribution:** 3 good

**Summary:**

This paper studies the problem of beneficial smoothing and oversmoothing. The authors show that a finite number of mean aggregation steps improve the learning performance of linear GNNs before oversmoothing occurs in examples of regression and classification. Numerical experiments demonstrate their claims.

**Questions:**

See weaknesses.

**Limitations:**

Yes.

**Strengths And Weaknesses:**

Strengths:
1. The paper provides the first theoretical study that models the benefits of finite smoothing (mean aggregation) before oversmoothing occurs.
2. The paper is well organized and easy to follow.


Weaknesses:
1. The authors may want to conduct experiments on some large-scale datasets, such as OGB [1].
2. The authors may want to analyze some non-linear GNNs, which are more useful and powerful for large-scale datasets than linear GNNs.
3. The authors may want to provide approaches to efficiently compute or estimate the optimal smoothing point, which is useful in practice.
[1] Hu, Weihua, et al. "Open graph benchmark: Datasets for machine learning on graphs." Advances in neural information processing systems 2020.

---

> ### Author Response · Authors · 2022-07-27
> **First rebuttal**
>
> We thank the reviewer for their feedback.
>
> We try to answer the questions raised by the reviewer below.
> - This is an interesting point. We conducted small illustrative experiments only, to observe the expected shrinking phenomenon along principal directions, but it is true that large-scale datasets may exhibit different behaviors. We will conduct more experiments, and if any interesting observation is made (significantly different behavior), put them in the appendix.
> - This is an outstanding open question, and at the time, we are not able to perform this analysis for non-linear GNNs, beyond the existing ones for oversmoothing only. It is the topic of future investigations.
> - One smoothing step is linear in the number of edges of the graph, so very efficient for sparse graph. In real-world scenarii, in the absence of prior knowledge, we do not believe that there is a more efficient way of estimating the optimal $k^\star$ than actually performing the smoothing steps one after another and observing the empirical results, at least in the context of Linear Ridge Regression, which is also fast to compute. However, we do believe that various quantities can be estimed during this process, eg, which directions are actually shrinked by the graph structure and how much. This knowledge could be exploited in designing more complex architectures with appropriate implicit bias without having to test every depth $k$. We will add a more detailed discussion on this point in the paper.

---

### Official Review · Reviewer_k824 · 2022-07-11

**Rating:** 8
**Confidence:** 4
**Soundness:** 4 excellent
**Presentation:** 4 excellent
**Contribution:** 4 excellent

**Summary:**

This work studies the phenomenon of oversmoothing in graphical neural networks and asks the question: since message passing averages information and thus loses local information, why is it ever helpful in estimation.  The authors consider a linear model in which the data distribution is Gaussian and message passing and aggregation are linear, so the model can be exactly solved.  It is shown that k rounds of message passing takes a data covariance \Sigma to (Id + \Sigma^-1)^{-2k} \Sigma, a transformation which shrinks small eigenvalues.  If these are noise directions then this is beneficial.

**Questions:**

I found the discussion in section 4.2 particularly interesting and it raises a question.  As the authors say, the result depends on parameters of the random graph model, in particular the covariance in equation 6.  It is taken to be 1 here and the general dependence is easy to put in.  But if one wants to apply the result to an empirical graph and observations, one needs to know how to estimate this parameter, as the authors comment in the conclusions.  Equivalently, what determines which are the "small" eigenvalues which get smoothed versus the "large" ones which do not ?


**Limitations:**

Technical results with no immediate societal impact.

**Strengths And Weaknesses:**

Strength: a simple, clear and instructive analysis bearing on a question of much current interest.

Weakness: more could be said about the simple model, for example about the question raised below.

---

> ### Author Response · Authors · 2022-07-27
> **First rebuttal**
>
> We thank the reviewer for their feedback.
>
> Two questions were raised by the reviewer, concerning the evaluation of the graph parameters in real-world situations, and what constitute "large" and "small" eigenvalues in a regression context.
>
> - concerning the second question (and part of the first), the covariance of the kernel has been taken to be identity here for simplicity, but it could have been any covariance matrix. The same computations would hold, with an additional "skewness" of the entire space (as if the data were deformed by the inverse of the covariance matrix of the kernel). That said, this covariance is also the threshold that determines "large" and "small" eigenvalues: the "fast vs slow shrinking" intuition is valid when eigenvalues are respectively small or large before that of the kernel.
>
> - it is true that the situation presented here is idealized, and exploiting the intuitions derived here in real-world scenarii would ask for a number of approximate estimation steps. Edge connections are obviously rarely as "smooth" as a pure gaussian kernel, but they may "act" like it in terms of smoothing various directions in the data. One could imagine performing a number of smoothing steps to evaluate how much each directions is shrinked (for instance in a PCA decomposition like the Cora illustration in Fig 1), that would give a rough idea on how much the graph structure actually act like a smoothing kernel with a particular covariance structure. This knowledge could be in turn exploited to induce various implicit bias in the construction of other architectures. We will add a more detailed discussion in the paper.

---

### Meta-Review · Area_Chair_oJEG · 2022-08-26

**Recommendation:** Accept
**Confidence:** Certain

**Metareview:**

The majority of reviewers are in favor of accepting * Not too little, not too much: a theoretical analysis of graph (over)smoothing* .  The reviewers were impressed in general by this theoretical analysis of finite-step mean aggregation smoothing in linear GNNs which is well-supported by their simulation results.  The paper's model gives evidence of both the value of smoothing but demonstrates that a threshold for over smoothing exists.  In general the novelty of the analysis of  a phenomena relevant to the community and the quality presentation leads me to recommend that this paper be accepted.

**Award:**

No

---

### Decision · Program_Chairs · 2022-09-14

Accept